# THE SHAPE AND SIMPLICITY BIASES OF ADVERSARIALLY ROBUST IMAGENET-TRAINED CNNS

## ABSTRACT

Adversarial training has been the topic of dozens of studies and a leading method for defending against adversarial attacks. Yet, it remains largely unknown (a) how adversarially-robust ImageNet classifiers (R classifiers) generalize to out-of-distribution examples; and (b) how their generalization capability relates to their hidden representations. In this paper, we perform a thorough, systematic study to answer these two questions across AlexNet, GoogLeNet, and ResNet-50 architectures. We found that while standard ImageNet classifiers have a strong texture bias, their R counterparts rely heavily on shapes. Remarkably, adversarial training induces three simplicity biases into hidden neurons in the process of "robustifying" the network. That is, each convolutional neuron in R networks often changes to detecting (1) pixel-wise smoother patterns i.e. a mechanism that blocks high-frequency noise from passing through the network; (2) more lower-level features i.e. textures and colors (instead of objects); and (3) fewer types of inputs. Our findings reveal the interesting mechanisms that made networks more adversarially robust and also explain some recent findings e.g. why R networks benefit from much larger capacity (Xie & Yuille, 2020) and can act as a strong image prior in image synthesis (Santurkar et al., 2019).

## 1 INTRODUCTION

Given excellent test-set performance, deep neural networks often fail to generalize to out-of-distribution (OOD) examples (Nguyen et al., 2015) including "adversarial examples", i.e. modified inputs that are imperceptibly different from the real data but change predicted labels entirely (Szegedy et al., 2014). Importantly, adversarial examples can transfer between models and cause unseen, all machine learning (ML) models to misbehave (Papernot et al., 2017), threatening the security and reliability of ML applications (Akhtar & Mian, 2018). Adversarial training—teaching a classifier to correctly label adversarial examples (instead of real data)—has been a leading method in defending against adversarial attacks and the most effective defense in ICLR 2018 (Athalye et al., 2018). Besides improved performance on adversarial examples, test-set accuracy can also be improved, for some architectures, when real images are properly incorporated into adversarial training (Xie et al., 2020). It is therefore important to study how the standard adversarial training (by Madry et al. 2018) changes the hidden representations and generalization capabilities of neural networks.

On smaller datasets, Zhang & Zhu (2019) found that adversarially-robust networks (hereafter, R networks) rely heavily on shapes (instead of textures) to classify images. Intuitively, training on pixel-wise noisy images would encourage R networks to focus less on local statistics (e.g. textures) and instead harness global features (e.g. shapes) more. However, an important, open question is:

*Q1: On ImageNet, do R networks still prefer shapes over textures?*

It remains unknown whether such shape preference carries over to the large-scale ImageNet (Russakovsky et al., 2015), which often induces a large texture bias into networks (Geirhos et al., 2019) e.g. to separate ~150 four-legged species in ImageNet. Also, this shape-bias hypothesis suggested by Zhang & Zhu (2019) seems to contradict the recent findings that R networks on ImageNet act as a strong *texture* prior i.e. they can be successfully used for many image translation tasks without any extra image prior (Santurkar et al., 2019). The above discussion leads to a follow-up question:

***Q2:*** *If an R network has a stronger preference for shapes than standard ImageNet networks (hereafter, S networks), will it perform better on OOD distorted images?*

Networks trained to be more *shape*-biased can generalize better to many unseen ImageNet-C (Hendrycks & Dietterich, 2019) image corruptions than S networks, which have a strong *texture* bias (Brendel & Bethge, 2019). In contrast, there was also evidence that classifiers trained on one type of images often do not generalize well to others (Geirhos et al., 2018; Nguyen et al., 2015; Kang et al., 2019). Importantly, R networks often underperform S networks on original test sets (Tsipras et al., 2019) perhaps due to an inherent trade-off (Madry et al., 2018), a mismatch between real vs. adversarial distributions (Xie et al., 2020), or a limitation in architectures—AdvProp helps improving performance of EfficientNets but not ResNets (Xie et al., 2020).

Most previous work aimed at understanding the behaviors of R classifiers as a function but little is known about the internal characteristics of R networks and, furthermore, their connections to the shape bias and generalization performance. Here, we ask:

***Q3:*** *How did adversarial training change the hidden neural representations to make classifiers more shape-biased and adversarially robust?*

In this paper, we harness the common benchmarks in ML interpretability and neuroscience—cue-conflict (Geirhos et al., 2019), NetDissect (Bau et al., 2017), and ImageNet-C—to answer the three questions above via a systematic study across three different convolutional architectures—AlexNet (Krizhevsky et al., 2012), GoogLeNet (Szegedy et al., 2015), and ResNet-50 (He et al., 2016)—trained to perform image classification on the large-scale ImageNet dataset (Russakovsky et al., 2015). Our main findings include:[1]

1. R classifiers trained on ImageNet prefer shapes over textures ∼67% of the time (Sec. 3.1)—a stark contrast to the S classifiers, which use shapes at only ∼25%.

2. Consistent with the strong shape bias, R classifiers interestingly outperform S counterparts on texture-less, distorted images (stylized and silhouetted images) (Sec. 3.2.2).

3. Adversarial training makes R networks more robust by (1) blocking pixel-wise input noise via smooth filters (Sec. 3.3.1); (2) narrowing the input range that highly activates neurons to simpler patterns, effectively reducing the space of adversarial inputs (Sec. 3.3.2).

4. Units that detect texture patterns (according to NetDissect) are not only useful to texture-based recognition as expected but can be also highly useful to *shape*-based recognition (Sec. 3.4). By aligning NetDissect and cue-conflict frameworks, we found that hidden neurons in R networks are surprisingly neither strongly shape-biased nor texture-biased, but instead generalists that detect low-level features (Sec. 3.4).

## 2 NETWORKS AND DATASETS

**Networks** To understand the effects of adversarial training across a wide range of architectures, we compare each pair of S and R models while keeping their network architectures constant. That is, we conduct all experiments on two groups of classifiers: (a) standard AlexNet, GoogLeNet, & ResNet-50 (hereafter, ResNet) models pre-trained on the 1000-class 2012 ImageNet dataset; and (b) three adversarially-robust counterparts i.e. AlexNet-R, GoogLeNet-R, & ResNet-R which were trained via adversarial training (see below) (Madry et al., 2018).

**Training** A standard classifier with parameters $\theta$ was trained to minimize the cross-entropy loss $L$ over pairs of (training example $x$, ground-truth label $y$) drawn from the ImageNet training set $\mathcal{D}$:

$$\arg\min_{\theta} \mathbb{E}_{(x,y)\sim\mathcal{D}}\Big[L(\theta, x, y)\Big] \tag{1}$$

On the other hand, we trained each R classifier via Madry et al. (2018) adversarial training framework where each real example $x$ is changed by a perturbation $\Delta$:

$$\arg\min_{\theta} \mathbb{E}_{(x,y)\sim\mathcal{D}}\Big[\max_{\Delta\in\mathcal{P}} L(\theta, x + \Delta, y)\Big] \tag{2}$$

---

[1]All code and data will be available on github upon publication.

where $\mathcal{P}$ is the perturbation range (Madry et al., 2018), here, within an $L_2$ norm.

**Hyperparameters** The S models were downloaded from PyTorch model zoo (PyTorch, 2019). We trained all R models using the robustness library (Engstrom et al., 2019), using the same hyperparameters in Engstrom et al. (2020); Santurkar et al. (2019); Bansal et al. (2020). That is, adverarial examples were generated using Projected Gradient Descent (PGD) (Madry et al., 2018) with an $L_2$ norm constraint $\epsilon$ of 3, a step size of 0.5, and 7 PGD-attack steps. R models were trained using an SGD optimizer for 90 epochs with a momentum of 0.9, an initial learning rate of 0.1 (which is reduced 10 times every 30 epochs), a weight decay of $10^{-4}$, and a batch size of 256 on 4 Tesla-V100 GPU's.

Compared to the standard counterparts, R models have substantially higher adversarial accuracy but lower ImageNet validation-set accuracy (Table 1). To compute adversarial accuracy, we perturbed validation-set images with the same PGD attack settings as used in training.

Table 1: Top-1 accuracy (%) on 50K-image ImageNet validation-set and PGD adversarial examples.

|  | AlexNet | AlexNet-R | GoogLeNet | GoogLeNet-R | ResNet | ResNet-R |
|---|---|---|---|---|---|---|
| ImageNet | **56.52** | 39.83 | **69.78** | 43.57 | **76.13** | 57.90 |
| Adversarial | 0.18 | **22.27** | 0.08 | **31.23** | 0.35 | **36.11** |

**Correctly-labeled image subsets: ImageNet-CL** Following Bansal et al. (2020), to compare the behaviors of two networks of identical architectures on the same inputs, we tested them on the largest ImageNet validation subset (hereafter, ImageNet-CL) where both models have 100% accuracy. The sizes of the three subsets for three architectures—AlexNet, GoogLeNet, and ResNet—are respectively: 17,693, 24,581, and 27,343. On modified ImageNet images (e.g. ImageNet-C), we only tested each pair of networks on the modified images whose original versions exist in ImageNet-CL. That is, we wish to gain deeper insights into how networks behave on correctly-classified images, and then how their behaviors change when some input feature (e.g. textures or shapes) is modified.

## 3 EXPERIMENT AND RESULTS

### 3.1 DO IMAGENET ADVERSARIALLY ROBUST NETWORKS PREFER SHAPES OR TEXTURES?

It is important to know which type of feature a classifier uses when making decisions. While standard ImageNet networks often carry a strong texture bias (Geirhos et al., 2019), it is unknown whether their adversarially-robust counterparts would be heavily texture- or shape-biased. Here, we test this hypothesis by comparing S and R models on the well-known cue-conflict dataset (Geirhos et al., 2019). That is, we feed "stylized" images provided by Geirhos et al. (2019) that contain contradicting texture and shape cues (e.g. elephant skin on a cat silhouette) and count the times a model uses textures or shapes (i.e. outputting elephant or cat) when it makes a correct prediction.

**Experiment** Our procedure follows Geirhos et al. (2019). First, we excluded 80 images that do not have conflicting cues (e.g. cat textures on cat shapes) from their 1,280-image dataset. Each texture or shape cue belongs to one of 16 MS COCO (Caesar et al., 2018) coarse labels (e.g. cat or elephant). Second, we ran the networks on these images and converted their 1000-class probability vector outputs into 16-class probability vectors by taking the average over the probabilities of the fine-grained classes that are under the same COCO label. Third, we took only the images that each network correctly labels (i.e. into the texture or shape class), which ranges from 669 to 877 images (out of 1,200) for 6 networks and computed the texture and shape accuracies over 16 classes.

**Results** On average, over three architectures, R classifiers rely on shapes $\geq 67.08\%$ of the time i.e. $\sim2.7\times$ higher than 24.56% of the S models (Table 2). In other words, by replacing the real examples with adversarial examples, adversarial training causes the heavy texture bias of ImageNet classifiers (Geirhos et al., 2019; Brendel & Bethge, 2019) to drop substantially ($\sim2.7\times$).

### 3.2 DO ROBUST NETWORKS GENERALIZE TO UNSEEN TYPES OF DISTORTED IMAGES?

We have found that changing from standard training to adversarial training changes ImageNet classifiers entirely from texture-biased into shape-biased (Sec. 3.1). Furthermore, Geirhos et al. (2019)

Table 2: While standard classifiers rely heavily on textures, R classifiers rely heavily on shapes. The top-1 accuracy scores (%) are computed on the cue-conflict dataset by Geirhos et al. (2019).

|  | AlexNet | AlexNet-R | GoogLeNet | GoogLeNet-R | ResNet | ResNet-R |
|---|---|---|---|---|---|---|
| Texture | **73.61** | 34.67 | **74.91** | 34.43 | **77.79** | 29.63 |
| Shape | 26.39 | **65.32** | 25.08 | **65.56** | 22.20 | **70.36** |

found that some training regimes that encourage classifiers to focus more on *shape* can improve their performance on unseen image distortions. Therefore, it is interesting to test whether R models—a type of *shape*-biased classifiers— would generalize well to any OOD image types.

**ImageNet-C** We compare S and R networks on the ImageNet-C dataset which was designed to test model robustness on 15 common types of image corruptions (Fig. 1c), where several shape-biased classifiers were known to outperform S classifiers (Geirhos et al., 2019). Here, we tested each pair of S and R models on the ImageNet-C distorted images whose original versions were correctly labeled by both (i.e. in ImageNet-CL sets; Sec. 2).

**Results** R models show no generalization boost on ImageNet-C i.e. they performed on-par or worse than the S counterparts (Table 3c). This is consistent with the findings in Table 4 in Geirhos et al. (2019) that a stronger shape bias does not necessarily imply better generalizability.

To further understand the generalization capability of R models, we tested them on two controlled image types where either shape or texture cues are removed from the original, correctly-labeled ImageNet images. Note that when both shape and texture cues are present e.g. in cue-conflict images, R classifiers consistently prefer shape over texture i.e. a shape *bias*. However, this bias is *orthogonal* to the performance when only either texture or shape cues are present.

Table 3: R models often do not generalize well to common distorted images (c–d), which are outside their training distribution (b), but interestingly outperform S models on texture-less images (e–f). Here, we report top-1 accuracy scores (in %) on the transformed images whose original real versions were correctly labeled (a) by both S and R models. "ImageNet-C" column (c) shows the mean accuracy scores over all 15 distortion types (Hendrycks & Dietterich, 2019). "Scrambled" column (d) shows the mean accuracy scores over three patch-scrambling types (details in Fig. A1).

| Network | (a) Real | (b) Adv. | (c) ImageNet-C | (d) Scrambled | Shape-less (e) Stylized | Texture-less (f) Contour | (g) Silhouette |
|---|---|---|---|---|---|---|---|
| AlexNet | 100 | 0.18 | **21.10** | **34.59** | 6.31 | 20.08 | 7.72 |
| AlexNet-R | 100 | **22.27** | 20.19 | 16.92 | **9.11** | **35.25** | **9.30** |
| GoogLeNet | 100 | 0.08 | **37.90** | **49.74** | 13.74 | 43.48 | 10.17 |
| GoogLeNet-R | 100 | **31.23** | 26.23 | 31.15 | 12.54 | **44.55** | **24.12** |
| ResNet | 100 | 0.35 | **39.27** | **58.04** | 10.68 | 16.96 | 3.95 |
| ResNet-R | 100 | **36.11** | 31.13 | 34.46 | **15.62** | **53.89** | **22.30** |

(a) Real    (b) Adversarial  (c) ImageNet-C  (d) Scrambled    (e) Stylized    (f) B&W    (g) Silhouette

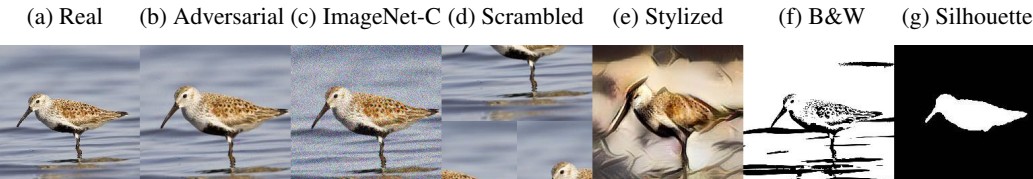

Figure 1: Example distorted images (b–g). We show an example Gaussian-noise-added image (c) out of all 15 ImageNet-C types. See Fig. A4 for more examples.

### 3.2.1 PERFORMANCE ON SHAPE-LESS, TEXTURE-PRESERVING IMAGES

We created shape-less images by dividing each ImageNet-CL image into a grid of $p \times p$ even patches where $p \in \{2, 4, 8\}$ and re-combining them randomly into a new "scrambled" version (Fig. 1d). On average, over three grid types, we observed a larger accuracy drop in R models compared to S models, ranging from $1.6\times$ to $2.04\times$ lower accuracy (Table 3d). That is, R model performance drops substantially when object shapes are removed—another evidence for their reliance on shapes. Compare predictions of ResNet vs. ResNet-R for scrambled images in Fig. A6. Remarkably, ResNet accuracy only drops from 100% to 94.77% on the $2 \times 2$ scrambled images (Fig. A1).

### 3.2.2 PERFORMANCE ON TEXTURE-LESS, SHAPE-PRESERVING IMAGES

Following Geirhos et al. (2019), we tested R models on three types of texture-less images where the texture is increasingly removed: (1) stylized ImageNet images where textures are randomly modified; (2) binary, black-and-white, i.e. B&W, images (Fig. 1f); and (3) silhouette images where the texture information is completely removed (Fig. 1e, g).

**Stylized ImageNet**   To construct a set of stylized ImageNet images (see Fig. 1e), we took all ImageNet-CL images (Sec. 2) and changed their textures via a stylization procedure in Geirhos et al. (2019), which harnesses the style transfer technique (Gatys et al., 2016) to apply a random style to each ImageNet "content" image.

**B&W images**   For all ImageNet-CL images, we used the same process described in Geirhos et al. (2019) to generate silhouettes, but we did not manually select and modify the images. We used the ImageMagick command-line tool (ImageMagick) to binarize ImageNet images into B&W images via the following steps:

1. **convert** image.jpeg image.bmp
2. **potrace - -svg** image.bmp **-o** image.svg
3. **rsvg-convert** image.svg > image.jpeg

**Silhouette**   For all ImageNet-CL images, we obtained their segmentation maps via a PyTorch DeepLab-v2 model (Chen et al., 2017) pre-trained on MS COCO-Stuff. We used the ImageNet-CL images that belong to a set of 16 COCO coarse classes in Geirhos et al. (2019) (e.g. bird, bicycle, airplane, etc.). When evaluating classifiers, an image is considered correctly labeled if its ImageNet predicted label is a subclass of the correct class among the 16 COCO classes (Fig. 1f; mapping sandpiper $\rightarrow$ bird).

**Results**   On all three texture-less sets, R models consistently outperformed their S counterparts (Table 3e–g)—a remarkable generalization capability, especially on B&W and silhouette images where all texture information is mostly removed.

## 3.3 HOW DOES ADVERSARIAL TRAINING MAKE NETWORKS MORE ROBUST?

What internal mechanisms help R networks become more robust? Here, we shed light into this question by analyzing R networks at the weight (Sec. 3.3.1) and neuron (Sec. 3.3.2) levels.

### 3.3.1 WEIGHT LEVEL: SMOOTH FILTERS TO BLOCK PIXEL-WISE NOISE

Consistent with Yin et al. (2019); Gilmer et al. (2019), we observed that AlexNet-R substantially outperforms AlexNet not only on adversarial examples but also several types of high-frequency image types (e.g. additive noise) in ImageNet-C (Table A1).

**Smoother filters**   To explain this phenomenon, we visualized the weights of all 64 conv1 filters ($11 \times 11 \times 3$), in both AlexNet and AlexNet-R, as RGB images. We compare each AlexNet conv1 filter with its nearest conv1 filter (via Spearman rank correlation) in AlexNet-R. Remarkably, R filters appear qualitatively much smoother than their counterparts (Fig. 2a). The R filter bank is also less diverse e.g. R edge detectors are often black-and-white in contrast to the colorful AlexNet edges (Fig. 2b). A similar contrast was also seen for the GoogLeNet and ResNet models (Fig. A3).

We also quantify the smoothness, in total variation (TV), of the filters of all 6 models (Table. 4) and found that, on average, the filters in R networks are much smoother. For example, the mean TV of

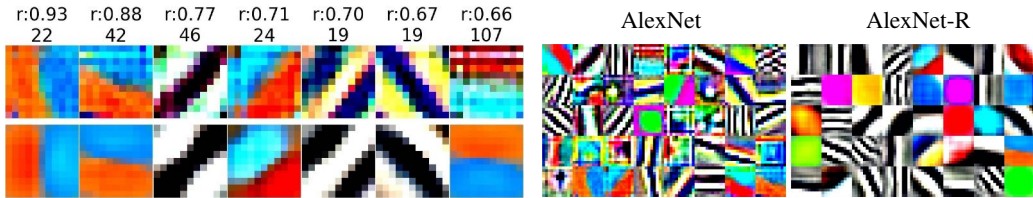

| r:0.93 | r:0.88 | r:0.77 | r:0.71 | r:0.70 | r:0.67 | r:0.66 |
| 22 | 42 | 46 | 24 | 19 | 19 | 107 |

(a) Standard filters (top) & matching R filters (bottom)    (b) 40 conv1 filters in AlexNet and AlexNet-R

Figure 2: **Left:** For each AlexNet conv1 filter (top row), we show the highest-correlated filter in AlexNet-R (bottom row), their Spearman rank correlation (e.g. r: 0.93) and the Total Variation (TV) difference (e.g. 22) between the top kernel and the bottom. Here, the TV differences are all positive i.e. AlexNet filters have higher TV. **Right:** conv1 filters of AlexNet-R are smoother and less diverse than the counterparts. Similar two plots for all 64 conv1 filters are in Figs. A2 & A3.

AlexNet-R is about 2 times smaller than AlexNet. Also, in lower layers, the filters in R classifiers are consistently 2 to 3 times smoother (Fig. A27).

**Blocking pixel-wise noise**  We hypothesize that the smoothness of filters makes R classifiers more robust against noisy images. To test this hypothesis, we computed the total variation (TV) (Rudin et al., 1992) of the channels across 5 conv layers when feeding ImageNet-CL images and their noisy versions (Fig. 1c; ImageNet-C Level 1 additive noise $\sim N(0, 0.08)$) to S and R models.

At conv1, the smoothness of R activation maps remains almost unchanged before and after noise addition (Fig. 3a; yellow circles are on the diagonal line). In contrast, the conv1 filters in standard AlexNet allow Gaussian noise to pass through, yielding larger-TV channels (Fig. 3a; blue circles are mostly above the diagonal). That is, the smooth filters in **R models indeed can filter out pixel-wise Gaussian noise despite that R models were not explicitly trained on this image type**! Interestingly, Ford et al. (2019) finding that the reverse engineering also works: training with Gaussian noise can improve adversarial robustness.

In higher layers, it is intuitive that the pixel-wise noise added to the input image might not necessarily cause activation maps, in both S and R networks, to be noisy because higher-layered units detect more abstract concepts. However, interestingly, we still found that R channels to have consistently less mean TV (Fig. 3b–c). Our result suggests that most of the de-noising effects take place at lower layers (which contain generic features) instead of higher layers.

Table 4: Mean total variance of conv layers of 6 different models. Our observation shows that the total variance in early layer of R models are consistently lower than S model. See Fig. A27 for layer-wsie analysis.

|  | AlexNet | AlexNet-R | GoogLeNet | GoogLeNet-R | ResNet | ResNet-R |
|---|---|---|---|---|---|---|
| Mean TV | 110.20 | **63.59** | 36.53 | **22.79** | 18.35 | 19.96 |

### 3.3.2 NEURON LEVEL: ROBUST NEURONS PREFER LOWER-LEVEL AND FEWER INPUTS

Here, via NetDissect framework, we wish to characterize how adversarial training changed the hidden neurons in R networks to make R classifiers more adversarially robust.

**Network Dissection**  (hereafter, NetDissect) is a common framework for quantifying the functions of a neuron by computing the Intersection over Union (IoU) between each activation map (i.e. channels) and the human-annotated segmentation maps for the same input images. That is, each channel is given an IoU score per human-defined concept (e.g. dog or zigzagged) indicating its accuracy in detecting images of that concept. A channel is tested for its accuracy on all $\sim$1,400 concepts, which span across six coarse categories: object, part, scene, texture, color, and material (Bau et al., 2017) (c.f. Fig. A11 for example NetDissect images in texture and color concepts). Following Bau et al. (2017), we assign each channel $C$ a main functional label i.e. the concept that $C$ has the highest IoU with. In both S and R models, we ran NetDissect on all 1152, 5808, and 3904 channels from,

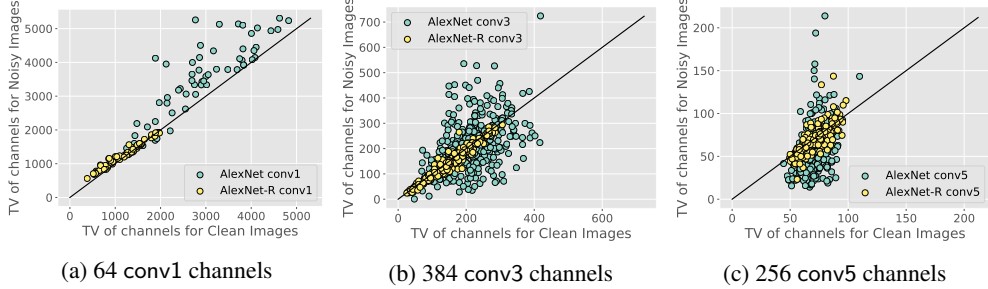

(a) 64 conv1 channels     (b) 384 conv3 channels     (c) 256 conv5 channels

Figure 3: In each subpanel, one point shows the mean Total Variation (TV) of one channel when running clean ImageNet-CL images and their noisy versions through AlexNet (teal) or AlexNet-R (yellow). R channels have similar TV before and after adding noise, suggesting that conv1 kernels filter out the added noise. In higher layers (conv3 and conv5), R channels are consistently more invariant to the input noise than S channels (yellow circles are clustered around the diagonal line while teal circles have higher variance). See Fig. A5 for the same scatter plot (a) for all five layers.

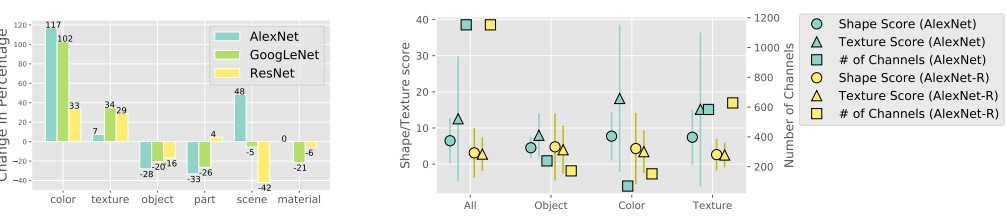

(a) Total channel increases (%) in R models

(b) Shape & Texture score of AlexNet & AlexNet-R

Figure 4: **Left:** For all three architectures, the numbers of color and texture detectors in R models increase, e.g. by 117% and 7%, respectively, for AlexNet, while the number of object units decreases by 28%. See Fig. A9 for layer wise plots for detectors in other category. **Right:** The average Shape (⬤) and Texture (▲) scores over all channels in the entire network ("All") or in a NetDissect category ("Object", "Color", and "Texture"). While AlexNet-R has more color and texture channels (▢ above ◼), these R channels are not heavily shape- or texture-biased. In contrast, the corresponding channels in AlexNet are heavily texture-biased (▲ is almost 2× of ⬤).

respectively, 5, 12, and 5 main convolutional layers (post-ReLU) of the AlexNet, GoogLeNet, and ResNet-50 architectures (c.f. Sec. A for more details of layers used).

**Shift to detecting more low-level features i.e. colors and textures** We found a consistent trend—adversarial training resulted in substantially more filters that detect colors and textures (i.e. in R models) in exchange for fewer object and part detectors. For example, throughout the same GoogLeNet architecture, we observed a 102% and a 34% increase of color and texture detectors, respectively, in the R model, but a 20% and a 26% fewer object and part detectors, compared to the S model (c.f. Fig. 4a). After adversarial training, ~11%, 15%, and 10% of all hidden neurons (in the tested layers) in AlexNet, GoogLeNet, and ResNet, respectively, shift their roles to detecting lower-level features (i.e. textures and colors) instead of higher-level features (Fig. A12). Across three architectures, the increases in texture and color channels are often larger in higher layers. While lower-layered units often learn more generic features, higher-layered units are more task-specific (Nguyen et al., 2016a), hence the largest functional shifts in higher layers.

We also compare the shape-biased ResNet-R with ResNet-SIN i.e. a ResNet-50 trained exclusively on stylized images (Geirhos et al., 2019), which also has a strong shape bias of 81.37%. [2] Interestingly, similar to ResNet-R, ResNet-SIN also have more low-level feature detectors (colors and textures) and fewer high-level feature detectors (objects and parts) than the vanilla ResNet (Fig. A28).

**Shift to detecting simpler objects** Analyzing the concepts in the object category where we observed largest changes in channel count, we found evidence that neurons change from detecting

---

[2] model_A in `https://github.com/rgeirhos/texture-vs-shape/`

complex to simpler objects. That is, for each NetDissect concept, we computed the difference in the numbers of channels between the S and R model. In the same object category, AlexNet-R model has substantially fewer channels detecting complex concepts e.g. $-30$ dog, $-13$ cat, and $-11$ person detectors (Fig. A8b; rightmost columns), compared to the standard network. In contrast, the R model has more channels detecting simpler concepts, e.g. $+40$ sky and $+12$ ceiling channels (Fig. A8b; leftmost columns). The top-49 images that highest-activated R units across five conv layers also show their strong preference for simpler backgrounds and objects (Figs. A15–A19).

**Shift to detecting fewer unique concepts**  The previous sections have revealed that neurons in R models often prefer images that are pixel-wise smoother (Sec. 3.3.1) and of lower-level features (Sec. 3.3.2), compared to S neurons. Another important property of the complexity of the function computed at each neuron is the diversity of types of inputs detected by the neuron (Nguyen et al., 2016b; 2019). Here, we compare the diversity score of NetDissect concepts detected by units in S and R networks. For each channel $C$, we calculated a diversity score i.e. the number of unique concepts that $C$ detects with an IoU score $\geq 0.01$.

Interestingly, on average, an R unit fires for 1.16 times fewer unique concepts than an S unit (22.43 vs. 26.07; c.f. Fig. A10a). Similar trends were observed in ResNet (Fig. A10b). Qualitatively comparing the highest-activation training-set images by the highest-IoU channels in both networks, for the same most-frequent concepts (e.g. striped), often confirms a striking difference: R units prefer a less diverse set of inputs (Fig. A12). As R hidden units fire for fewer concepts, i.e. significantly fewer inputs, the space for adversarial inputs to cause R models to misbehave is strictly smaller.

### 3.4 WHICH NEURONS ARE IMPORTANT FOR SHAPE- OR TEXTURE-BASED RECOGNITION?

To understand how the changes in R hidden neurons (Sec. 3.3) relate to the shape bias of R classifiers (Sec. 3.1), here, we zero out every channel, one at a time, in S and R networks and measure the performance drop in recognizing shape and texture from cue-conflict images.

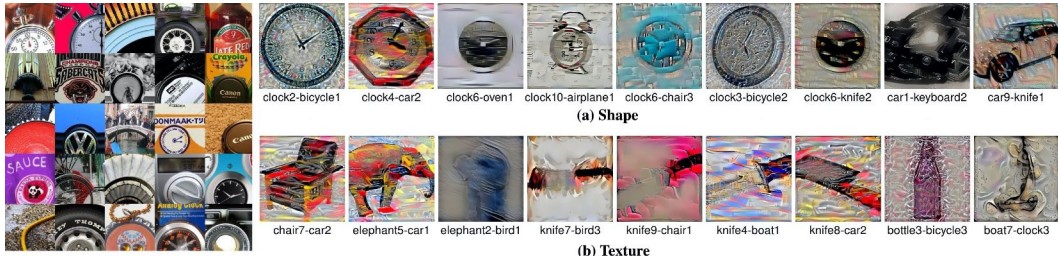

Figure 5: **Left:** Top-25 highest-activation images of the AlexNet unit $conv4_{19}$, which has a NetDissect label of spiralled under texture category. The unit prefer circular patterns e.g. car wheels and clock. **Right:** Example cue-conflict images originally labeled as shape (top) or texture (bottom) but that were given a different label after the unit is ablated. Qualitatively, the unit helps AlexNet detect clocks and cars using shapes (top row) and reddish pink cars, birds, chairs, and bicycles using textures (bottom row). The unit has Shape and Texture scores of 18 and 22. "clock2-bicycle1": the image has the shape of a clock and a texture of a bicycle. See Fig. A20 for a full version.

**Shape & Texture scores**  For each channel, we computed a Shape score i.e. the number of images originally correctly labeled into the shape class by the network but that, after the ablation, are labeled differently (examples in Fig 5a–b). Similarly, we computed a Texture score per channel. The Shape and Texture scores quantify the importance of a channel in classification using shapes or textures.

First, we found that the **channels labeled** texture **by NetDissect are not only important to texture-but also shape-based recognition**. That is, on average, zero-ing out these channels caused non-zero Texture and Shape scores (Fig. 4b; Texture ⬤ and △ are above 0). See Fig. 5 for an example of texture channels with high Shape and Texture scores.[3] This result sheds light into the fact that R networks consistently have more texture units (Fig. 4a) but are shape-biased (Sec. 3.1).

---

[3]Similar visualizations of some other neurons from both S and R networks are in Appendix Fig. A21–A26.

Second, the texture units are, as expected, highly texture-biased in AlexNet (Fig. 4b Texture; ▲ is almost 2× of ⬤). However, surprisingly, those texture **units in AlexNet-R are neither strongly shape-biased nor texture-biased** (Fig. 4b; Texture ◯ ≈ △). That is, across all three groups of the object, color, and texture, **R neurons appear mostly to be generalist, low-level feature detectors**. This generalist property might be a reason for why R networks are more effective in transfer learning than S networks (Salman et al., 2020).

Finally, the contrast above between the texture bias of S and R channels (Fig. 4b) reminds researchers that the single NetDissect label assigned to each neuron is not describing a full picture of what the neuron does and how it helps in downstream tasks. To the best of our knowledge, this is the first work to align the NetDissect and cue-conflict frameworks to study how individual neurons contribute to the generalizability and shape bias of the entire network.

## 4 DISCUSSION AND RELATED WORK

Deep neural networks tend to prioritize learning simple patterns that are common across the training set (Arpit et al., 2017). Furthermore, deep ReLU networks often prefer learning simple functions (Valle-Perez et al., 2019; De Palma et al., 2019), specifically low-frequency functions (Rahaman et al., 2019), which are more robust to random parameter perturbations. Along this direction, here, we have shown that R networks (1) have smoother weights (Sec. 3.3.1), (2) prefer even simpler and fewer inputs (Sec. 3.3.2) than standard deep networks—i.e. R networks represent even simpler functions. Such simplicity biases are consistent with the fact that gradient images of R networks are much smoother (Tsipras et al., 2019) and that R classifiers act as a strong image prior for image synthesis (Santurkar et al., 2019).

Each R neuron computing a more restricted function than an S neuron (Sec. 3.3.2) implies that R models would require more neurons to mimic a complex S network. This is consistent with recent findings that adversarial training requires a larger model capacity (Xie & Yuille, 2020).

While AdvProp did not yet show benefits on ResNet (Xie et al., 2020), it might be interesting future work to find out whether EfficientNets trained via AdvProp also have shape and simplicity biases. Furthermore, simplicity biases may be incorporated as regularizers into future training algorithms to improve model robustness. For example, encouraging filters to be smoother might improve robustness to high-frequency noise. Also aligned with our findings, Rozsa & Boult (2019) found that explicitly narrowing down the non-zero input regions of ReLUs can improve adversarial robustness.

We found that R networks heavily rely on shape cues in contrast to S networks. One may fuse an S network and a R network (two channels, one uses texture and one uses shape) into a single, more robust, interpretable ML model. That is, such model may (1) have better generalization on OOD data than S or R network alone and (2) enable an explanation to users on what features a network uses to label a given image.

Our study on how individual hidden neurons contribute to the R network shape preference (Sec. 3.4) revealed that texture-detector units are equally important to the texture-based and shape-based recognition. This is in contrast to a common hypothesis that texture detectors should be exclusively only useful to texture-biased recognition. Our surprising finding suggests that the categories of stimuli in the well-known Network Dissection (Bau et al., 2017) need to be re-labeled and also extended with low-frequency patterns e.g. single lines or silhouettes in order to more accurately quantify hidden representations.

## 5 CONCLUSION

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

## A    Convolutional layers used in Network Dissection analysis

For both standard and robust models, we ran NetDissect on 5 convolutional layers in AlexNet (Krizhevsky et al., 2012), 12 in GoogLeNet (Szegedy et al., 2015), and 5 in ResNet-50 architectures (He et al., 2016). For each layer, we use after-ReLU activations (if ReLU exists).

**AlexNet**    layers: conv1, conv2, conv3, conv4, conv5. Refer to these names in Krizhevsky et al. (2012).

**GoogLeNet**    layers: conv1, conv2, conv3, inception3a, inception3b, inception4a, inception4b, inception4c, inception4d, inception4e, inception5a, inception5b

Refer to these names in PyTorch code `https://github.com/pytorch/vision/blob/master/torchvision/models/googlenet.py#L83-L101`.

**ResNet-50**    layers: conv1, layer1, layer2, layer3, layer4

Refer to these names in PyTorch code `https://github.com/pytorch/vision/blob/master/torchvision/models/resnet.py#L145-L155`).

Table A1: Top-1 accuracy of 6 models (in %) on all 15 types of image corruptions in ImageNet-C (Hendrycks & Dietterich, 2019). On average over all 15 distortion types, R models underperform their standard counterparts.

|         |            | AlexNet | AlexNet-R | GoogLeNet | GoogLeNet-R | ResNet | ResNet-R |
|---------|------------|---------|-----------|-----------|-------------|--------|----------|
| Noise   | Gaussian   | 11.36   | 21.98     | 33.28     | 18.71       | 29.03  | 24.53    |
|         | Shot       | 10.55   | 21.35     | 31.01     | 17.86       | 26.97  | 23.92    |
|         | Impulse    | 7.74    | 19.68     | 24.54     | 15.30       | 23.55  | 21.07    |
| Blur    | Defocus    | 18.01   | 15.59     | 28.42     | 20.72       | 38.40  | 26.36    |
|         | Glass      | 17.37   | 17.91     | 23.91     | 29.02       | 26.78  | 34.29    |
|         | Motion     | 21.40   | 21.45     | 31.14     | 28.29       | 38.61  | 33.15    |
|         | Zoom       | 20.16   | 21.60     | 25.57     | 28.98       | 35.73  | 33.83    |
| Weather | Snow       | 13.32   | 12.25     | 32.66     | 21.36       | 33.19  | 25.83    |
|         | Frost      | 17.34   | 11.00     | 36.80     | 20.31       | 39.08  | 27.83    |
|         | Fog        | 18.07   | 1.83      | 42.80     | 3.48        | 46.17  | 5.65     |
|         | Brightness | 43.54   | 27.71     | 64.46     | 42.96       | 68.32  | 49.71    |
| Digital | Contrast   | 14.68   | 3.28      | 43.66     | 5.90        | 38.86  | 8.78     |
|         | Elastic    | 35.39   | 32.29     | 42.79     | 41.98       | 46.16  | 44.94    |
|         | Pixelate   | 28.22   | 36.33     | 54.86     | 48.11       | 44.49  | 52.62    |
|         | JPEG       | 39.35   | 38.65     | 52.57     | 50.44       | 53.80  | 54.37    |
| mean Accuracy |      | **21.10** | 20.19   | **37.90** | 26.23       | **39.27** | 31.13  |

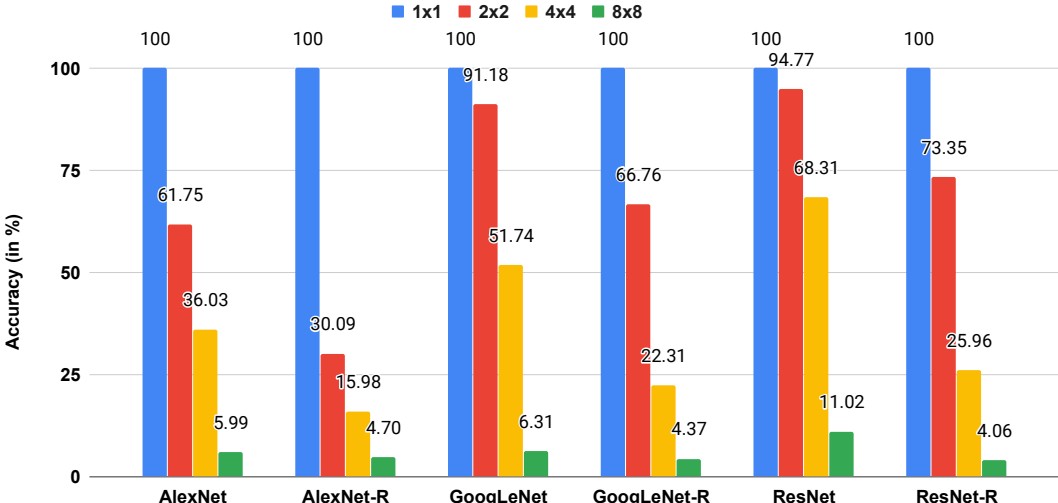

Figure A1: Standard models substantially outperform R models when tested on scrambled images due to their capability of recognizing images based on textures. See Fig. A6 for examples of scrambled images and their top-5 predictions from ResNet-R and ResNet (which achieves a remarkable accuracy of 94.77%).

Here, we report top-1 accuracy scores (in %) on the scrambled images whose original versions were correctly labeled by both standard and R classifiers (hence, the 100% for $1 \times 1$ blue bars).

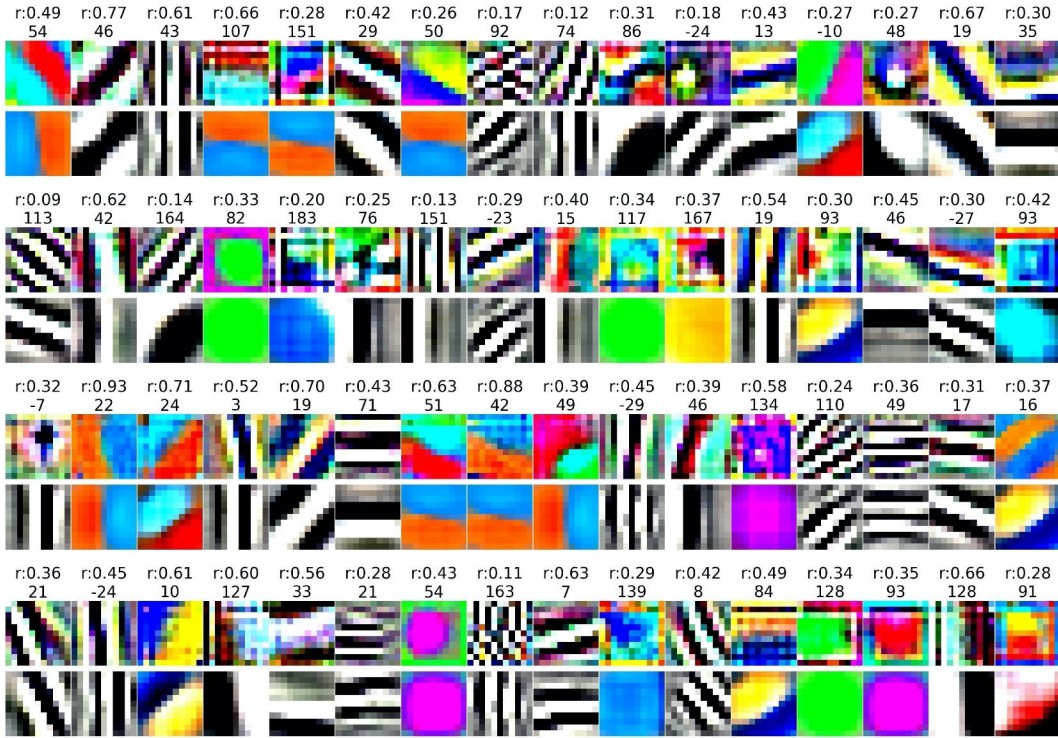

Figure A2: conv1 filters of AlexNet-R are smoother than the filters in standard AlexNet. In each column, we show an AlexNet filter conv1 filter and their nearest filter (bottom) from the AlexNet-R. Above each pair of filters are their Spearman rank correlation score (e.g. r: 0.36) and their total variation (TV) difference (i.e. smoothness differences). Standard AlexNet filters are mostly noisier than their nearest R filter (i.e. positive TV differences).

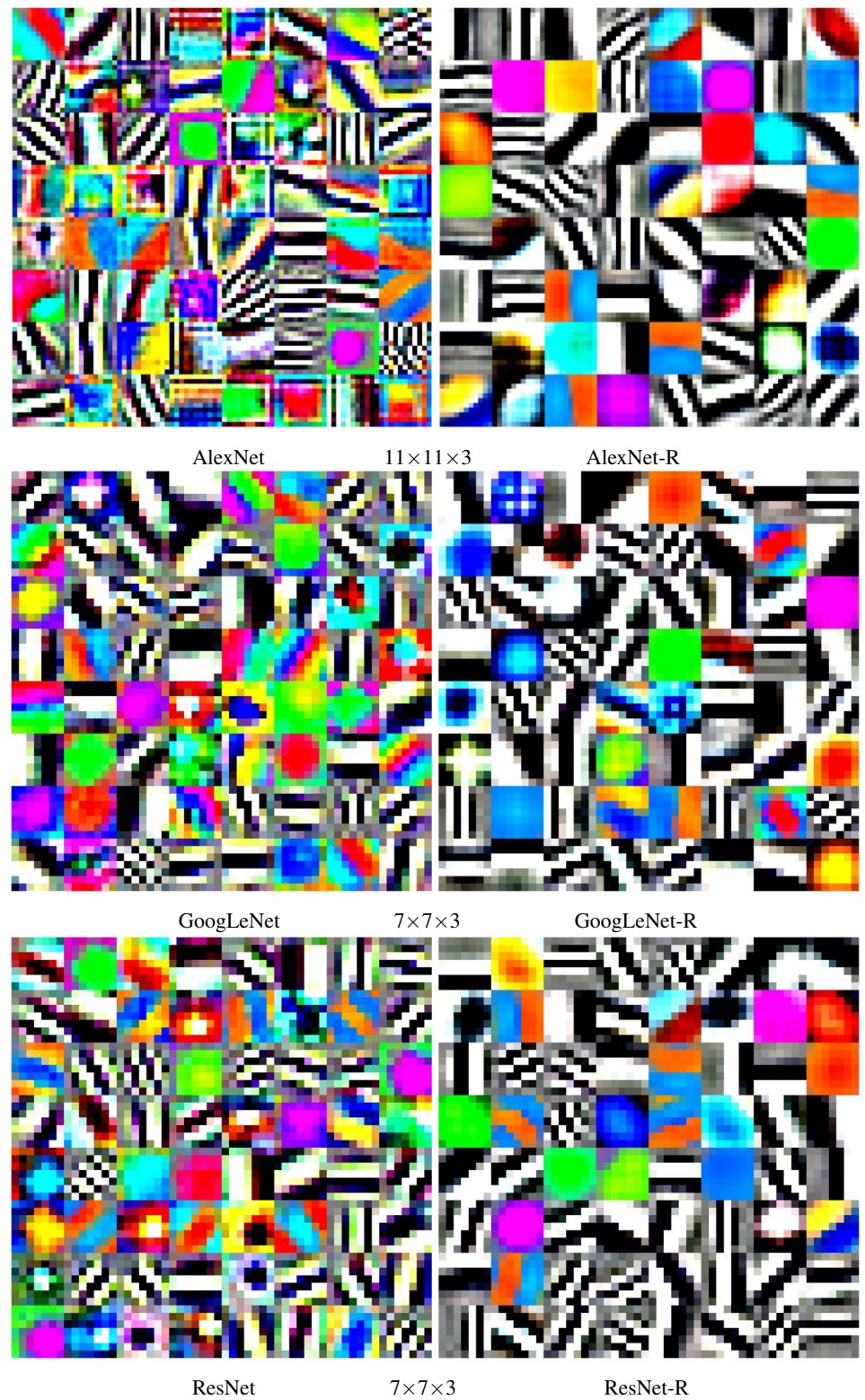

Figure A3: All 64 conv1 filters of in each standard network (left) and its counterpart (right). The filters of R models (right) are smoother and less diverse compared to those in standard models (left). Especially, the edge filters of standard networks are noisier and often contain multiple colors in them.

| (a) Real | (b) Scrambled | (c) Stylized | (d) Contour | (e) Silhouette |
|---|---|---|---|---|

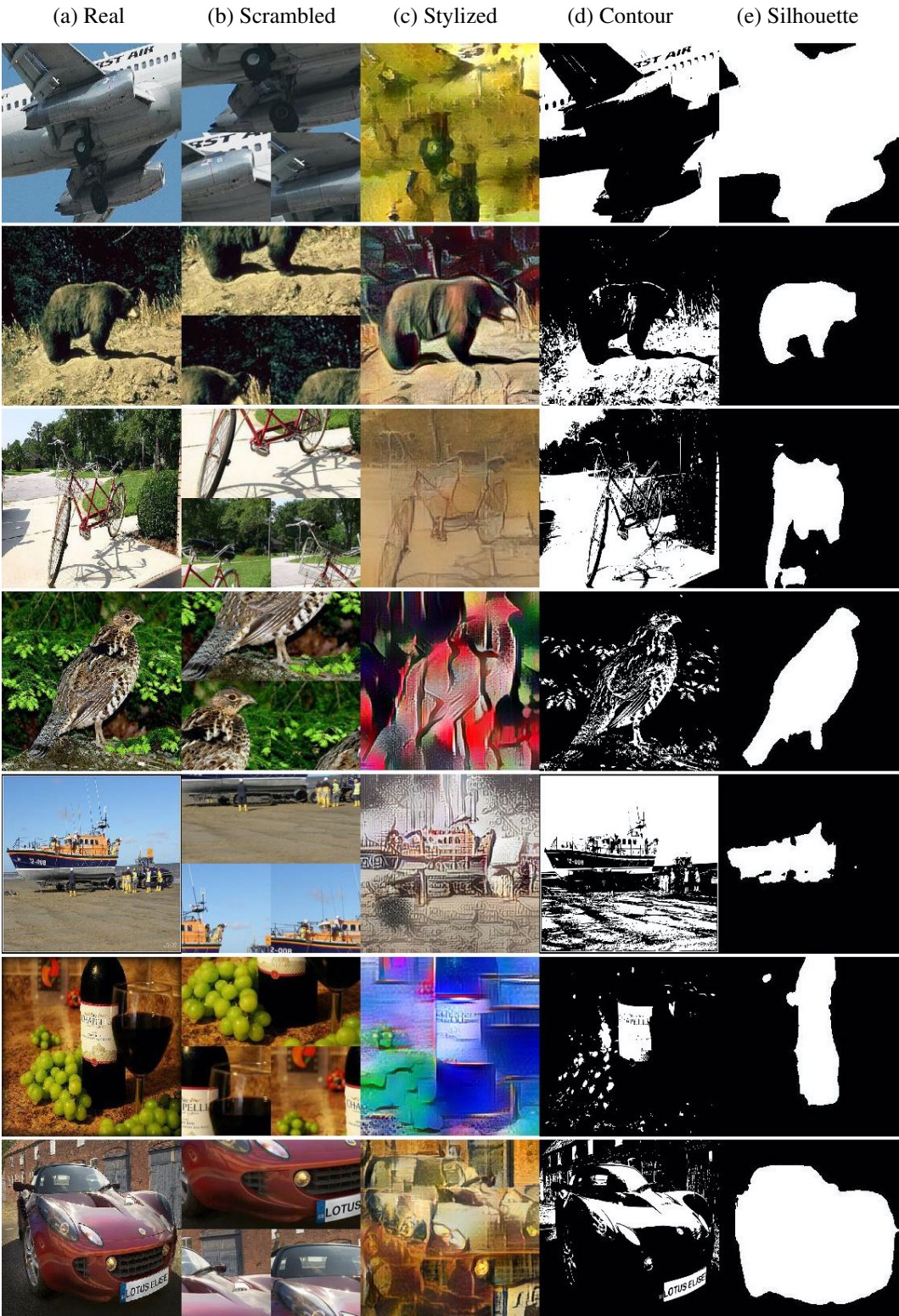

Figure A4: Applying different transformation that remove shape/texture on real images. We randomly show an example of 7 out of 16 COCO coarser classes. See Table 3 for classification accuracy scores on different images distortion dataset in 1000 classes(Except for Silhouette). *Note: Silhouette are validate in 16 COCO coarse classes.*

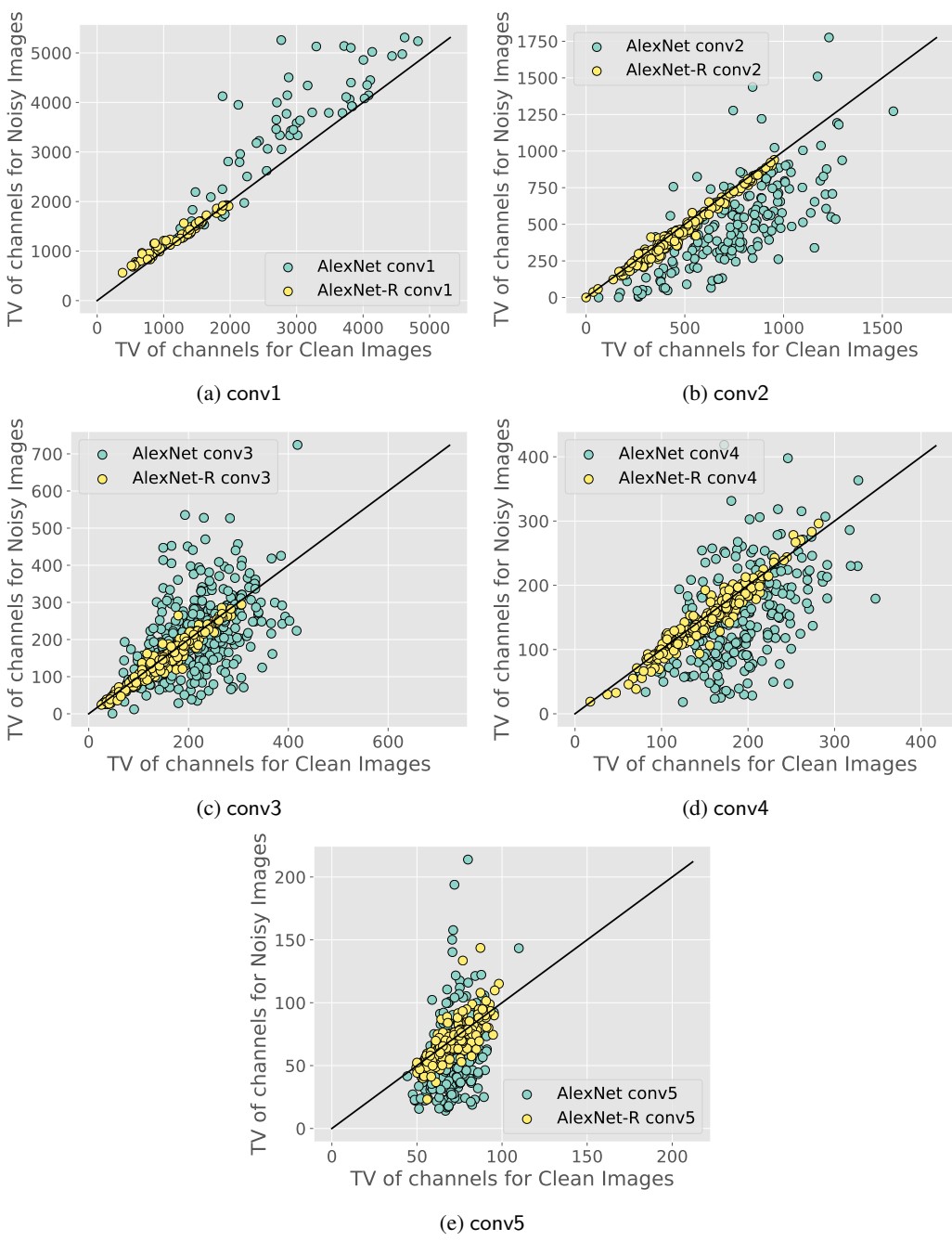

(a) conv1

(b) conv2

(c) conv3

(d) conv4

(e) conv5

Figure A5: Each point shows the Total Variation (TV) of the activation maps on clean and noisy images for an AlexNet or AlexNet-R channel. We observe a striking difference in conv1: The smoothness of R channels remains unchanged before and after noise addition, explaining their superior performance in classifying noisy images. While the channel smoothness differences (between two networks) are gradually smaller in higher layers, we still observe R channels are consistently smoother.

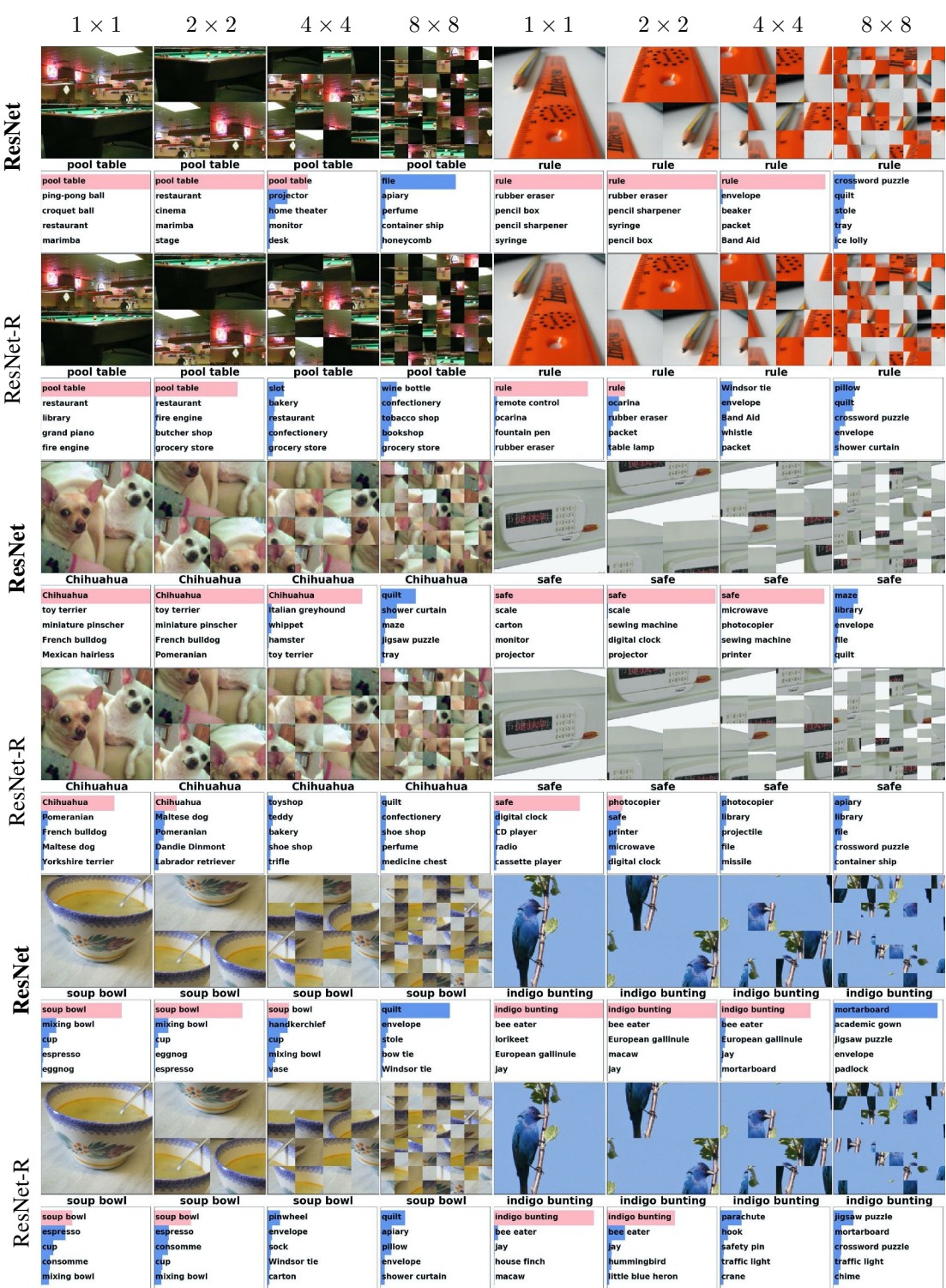

Figure A6: ResNet-R, on average across the three patch sizes, underperforms the standard ResNet model. Surprisingly, we observe that ResNet correctly classifies the image to their ground truth class even when the image is randomly shuffled into 16 patches, e.g., ResNet classifies the $4 \times 4$ case of rule, safe with $\sim 100\%$ confidence. The results are consistent with the strong texture bias of ResNet and shape bias of ResNet-R (described in Sec. 3.2.1).

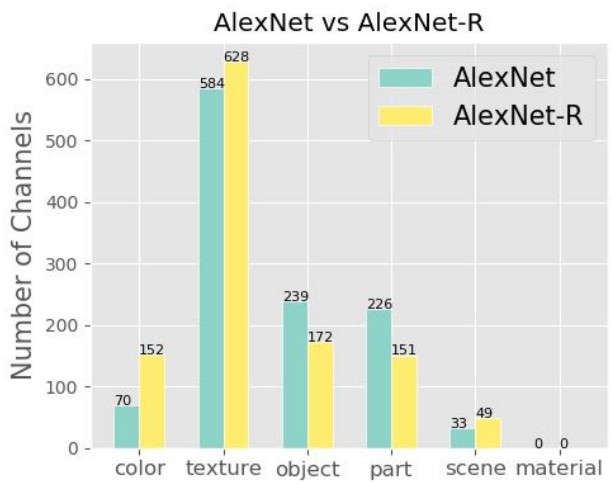

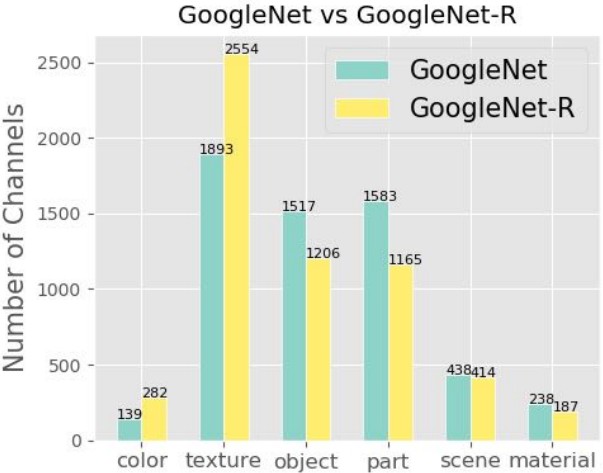

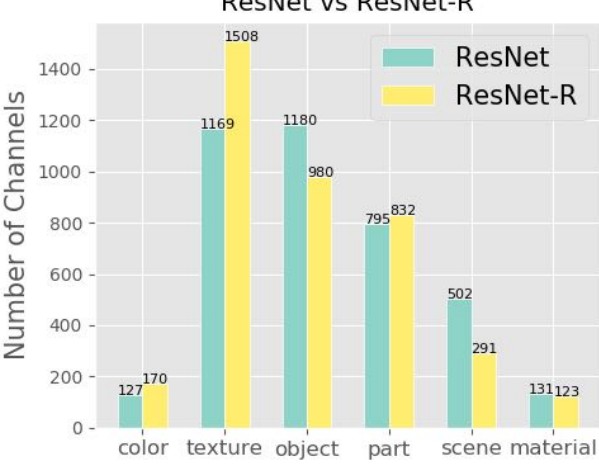

Figure A7: For each network, we show the number of channels in each of the 6 NetDissect categories (color, texture, etc) in Bau et al. (2017). Across all three architectures, R models consistently have more color and texture channels while substantially fewer object detectors.

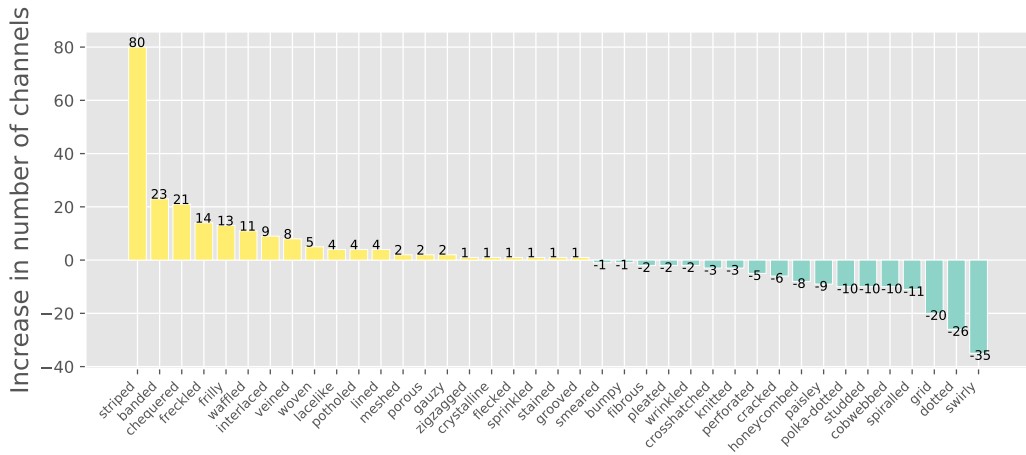

(a) Differences in texture channels between AlexNet and AlexNet-R

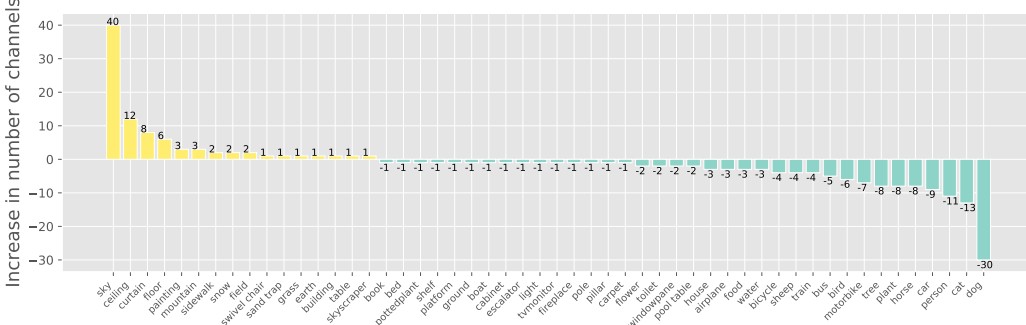

(b) Differences in object channels between AlexNet and AlexNet-R

Figure A8: In each bar plot, we column shows the difference in the number of channels (between AlexNet-R and AlexNet) for a given concept e.g. striped or banded. That is, yellow bars (i.e. positive numbers) show the count of channels that the R model has more than the standard network in the same concept. Vice versa, teal bars represent the concepts that R models have fewer channels. The NetDissect concept names are given in the x-axis.

**Top:** In the texture category, the R model has a lot more simple texture patterns e.g. striped and banded (see Fig. A11 for example patterns in these concepts).

**Bottom:** In the object category, AlexNet-R often prefers simpler-object detectors e.g. sky or ceiling (Fig. A8b; leftmost) while the standard network has more complex-objects detectors e.g. dog and cat (Fig. A8b; rightmost).

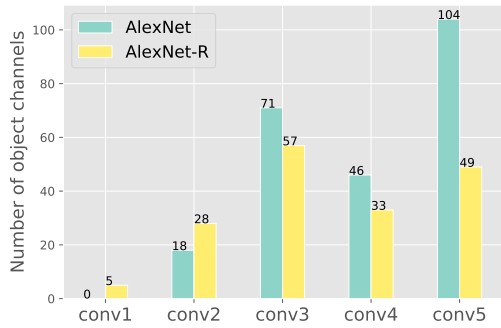 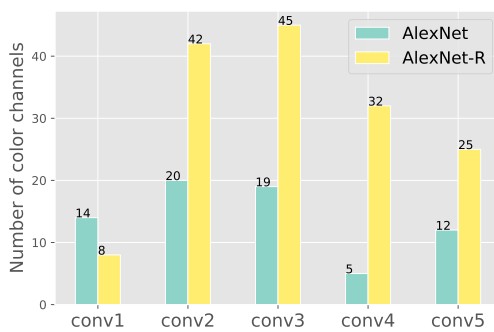

(a) Number of object detectors per AlexNet layer        (b) Number of color detectors per AlexNet layer

Figure A9: In higher layers (here, conv4 and conv5), AlexNet-R have fewer object detectors but more color detector units compared to standard AlexNet. The differences between the two networks increase as we go from lower to higher layers. Because both networks share an identical architecture, the plots here demonstrate a substantial shift in the functionality of the neurons as the result of adversarial training—detecting more colors and textures and fewer objects. Similar trends were also observed between standard and R models of GoogLeNet and ResNet-50 architectures.

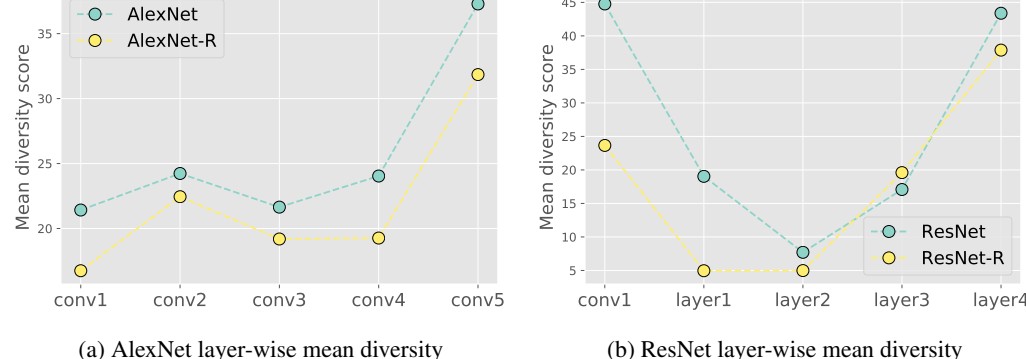

(a) AlexNet layer-wise mean diversity        (b) ResNet layer-wise mean diversity

Figure A10: In each plot, we show the mean diversity scores across all channels in each layer. Both AlexNet-R and ResNet-R consistently have channels with lower diversity scores (i.e. detecting fewer unique concepts) than the standard counterparts.

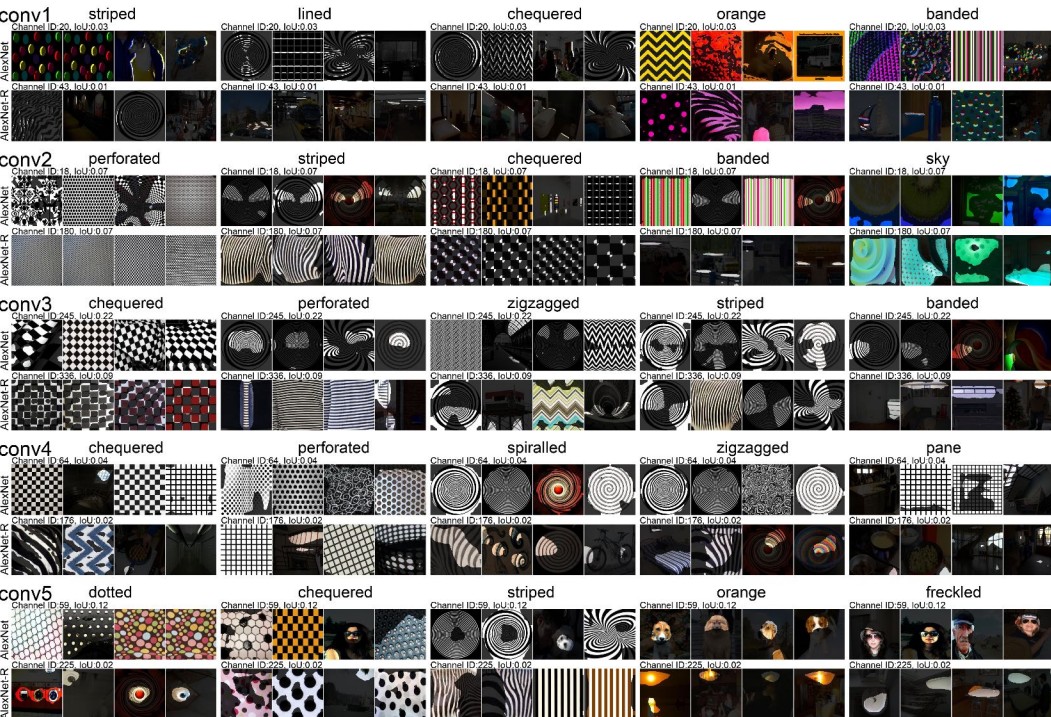

Figure A11: The NetDissect images preferred by the channels in the top-5 most important concepts in AlexNet (i.e. highest accuracy drop when zeroed out; see Sec. 3.3.2). For each concept, we show the highest-IoU channels.

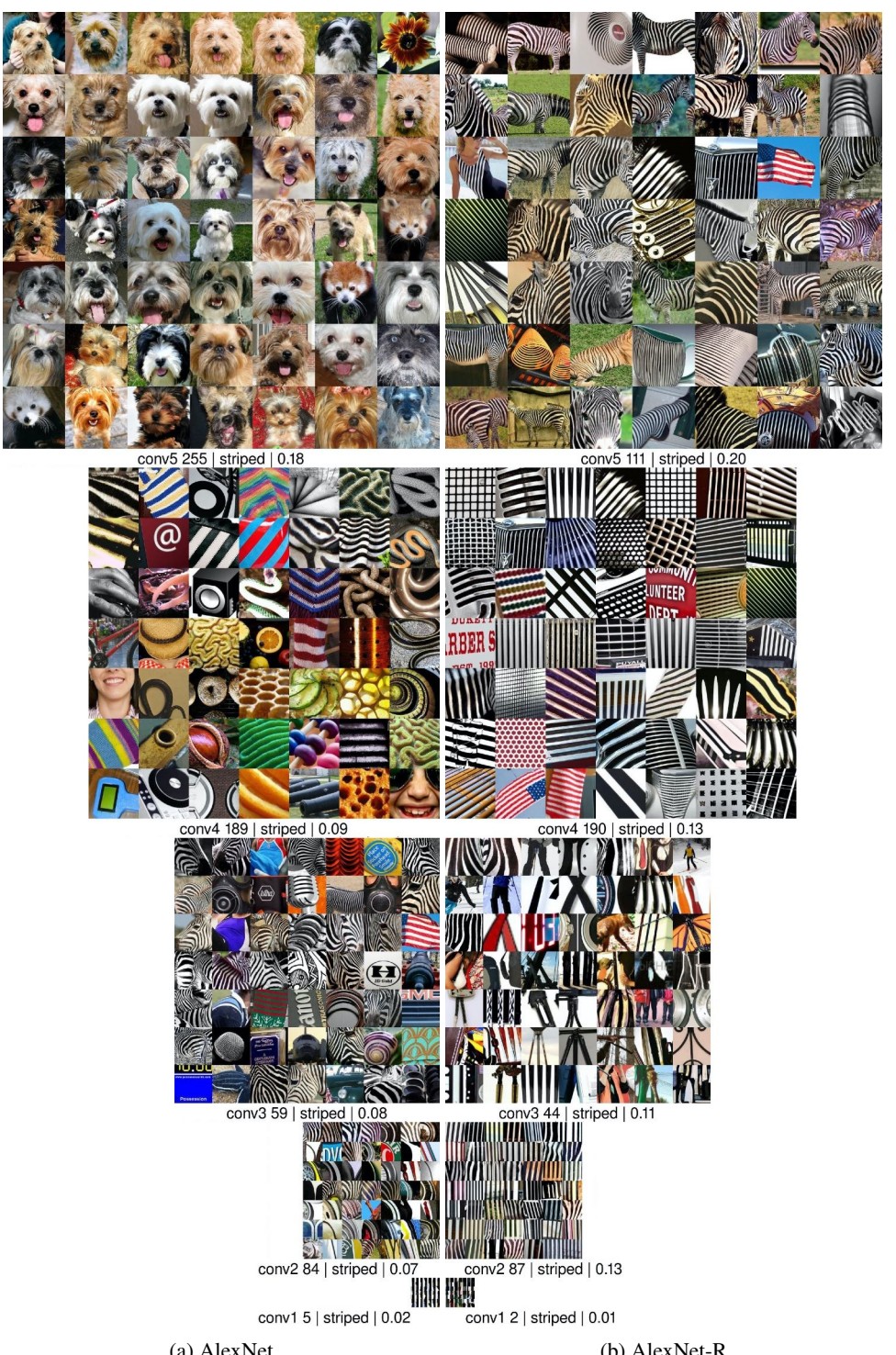

(a) AlexNet  (b) AlexNet-R

Figure A12: Each 7×7 grid shows the top-49 training-set images that highest activate the center unit in a channel. Each column shows five highest-IoU striped concept channels, each from one AlexNet's conv layer in their original resolutions. From top to bottom, AlexNet-R (b) consistently preferred striped patterns, i.e., edges (conv1), vertical bars (conv2), tools, to grids and zebra (conv5). In contrast, AlexNet striped images (a) are much more diverse, including curly patterns (conv4) and dog faces (conv5).

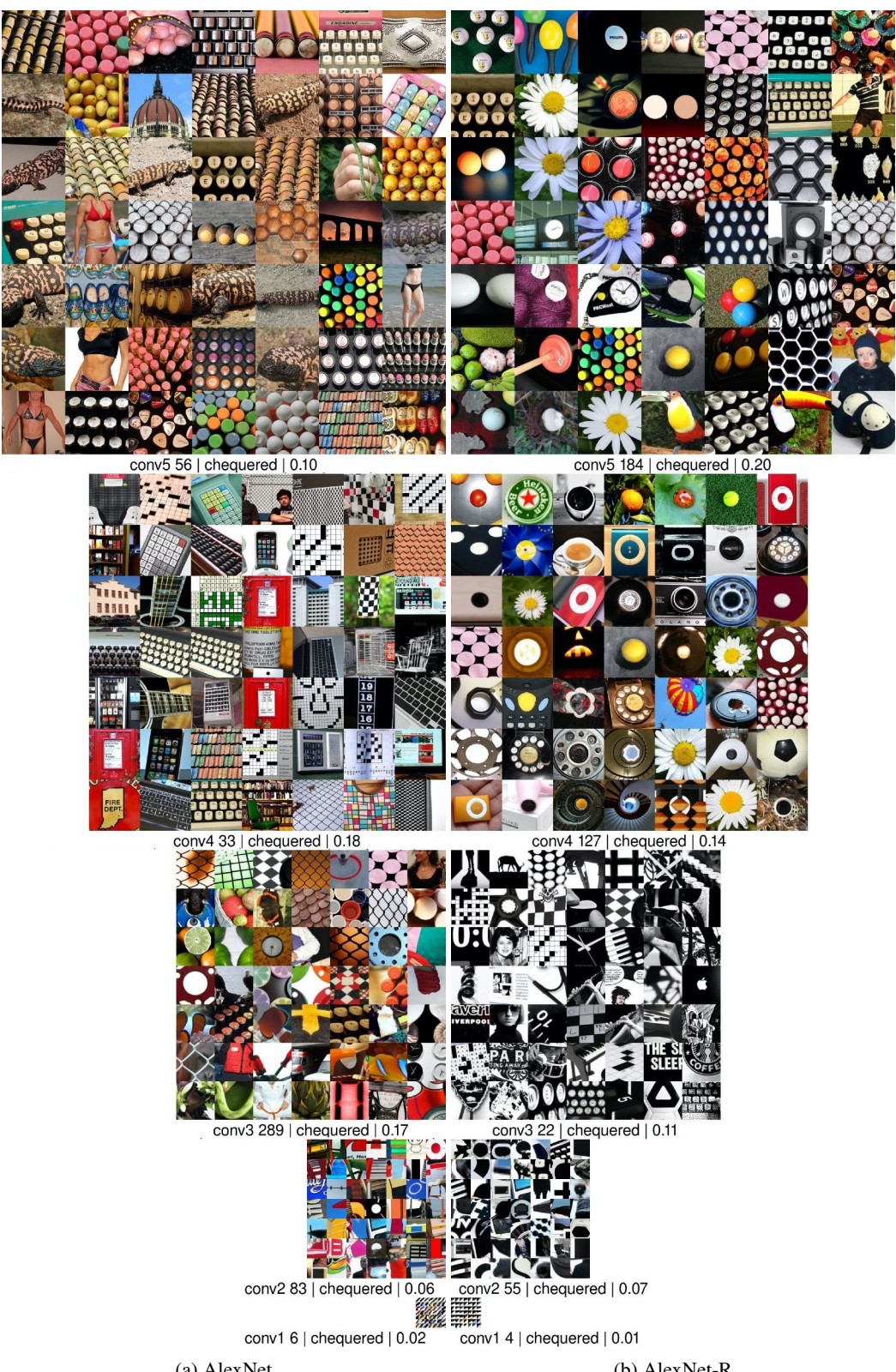

(a) AlexNet                                    (b) AlexNet-R

Figure A13: Same figure as Fig. A12 but for chequered concept. From top to bottom, AlexNet-R
(b) preferred consistently simple diagonal edges (conv1), chequered patterns (conv2), crosswords,
to dots and net patterns (conv5). In contrast, AlexNet images (a) are much more diverse, including
shelves (conv4) and gila monsters in (conv5). Remarkably, the R channels consistently prefer black-
and-white patterns in conv2 and conv3 while the standard channels prefer colorful images.

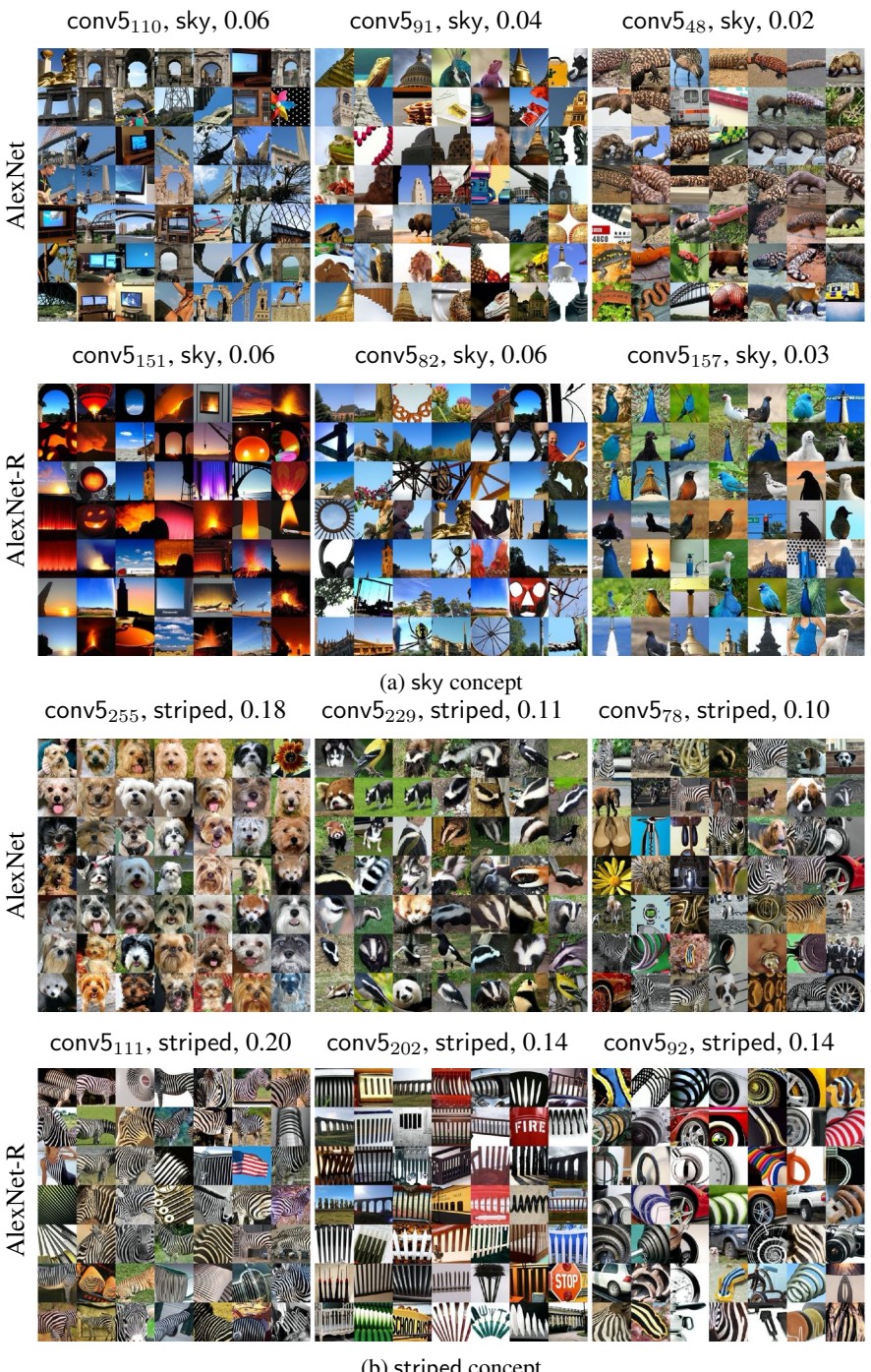

Figure A14: We show the top-49 highest-activation training-set images for the top 3 channels with highest IoU for the sky (a) and striped (b) concepts in the conv5 layer of AlexNet and AlexNet-R. The top-49 images for AlexNet-R nets align with the NetDissect concept labels whereas, for AlexNet, we observe diverse sets of preferred images of animals, such as, gilamonster and dog in sky (top panel) and striped concept (bottom panel) channels, respectively.

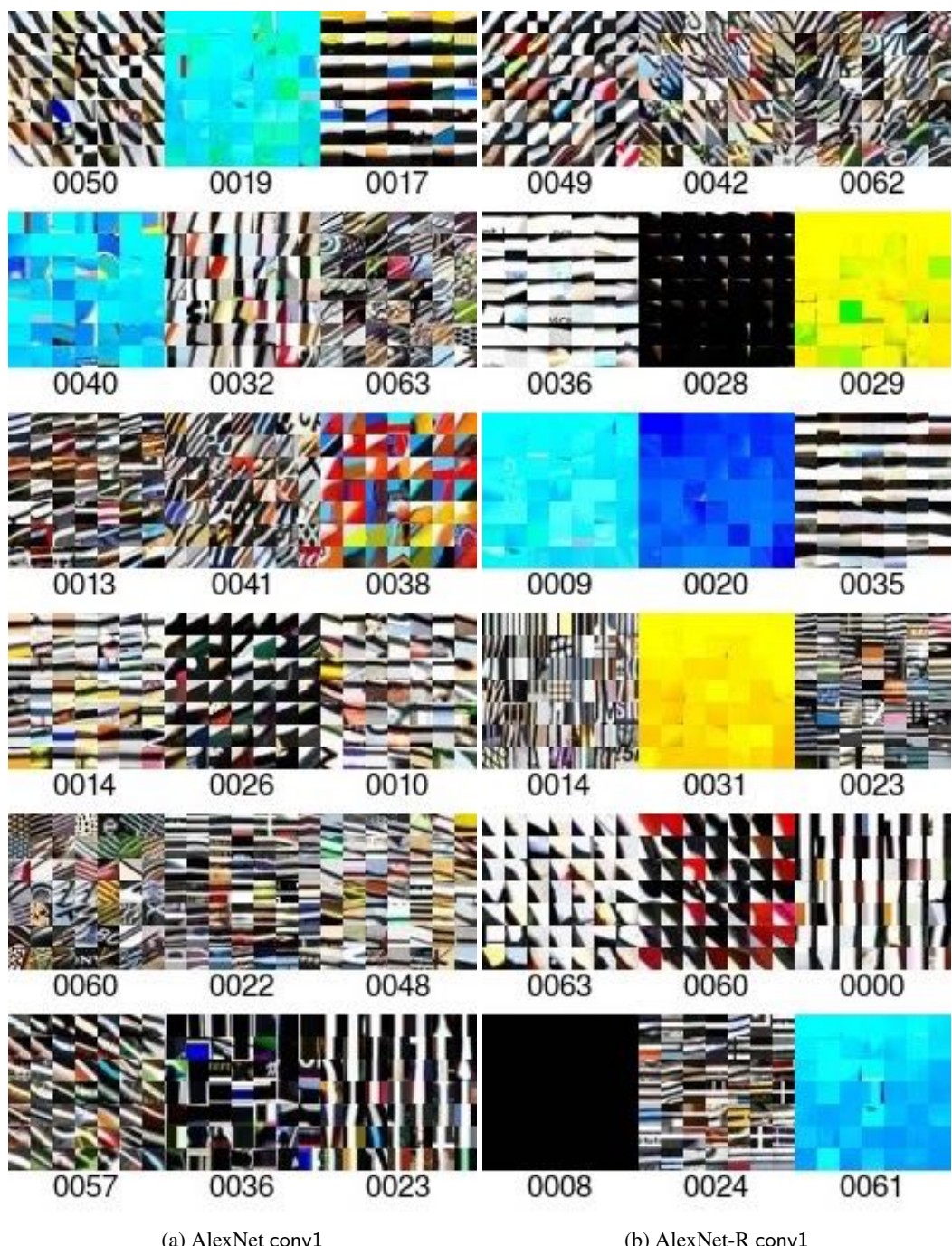

(a) AlexNet conv1          (b) AlexNet-R conv1

Figure A15: Each $7 \times 7$ grid shows the top-49 training-set images that highest activate the center unit in a channel for AlexNet (a) and AlexNet-R (b). A set of 20 channels were randomly sampled from the 64 channels in conv1 of the networks. The top-49 of AlexNet-R comprises of cleaner edge and color detector channels. For a complete set of 64 channels, refer to `https://drive.google.com/drive/folders/1eNmkxYT1nSBO-EWoeCkFDXAH60Rf8VrU?usp=sharing`.

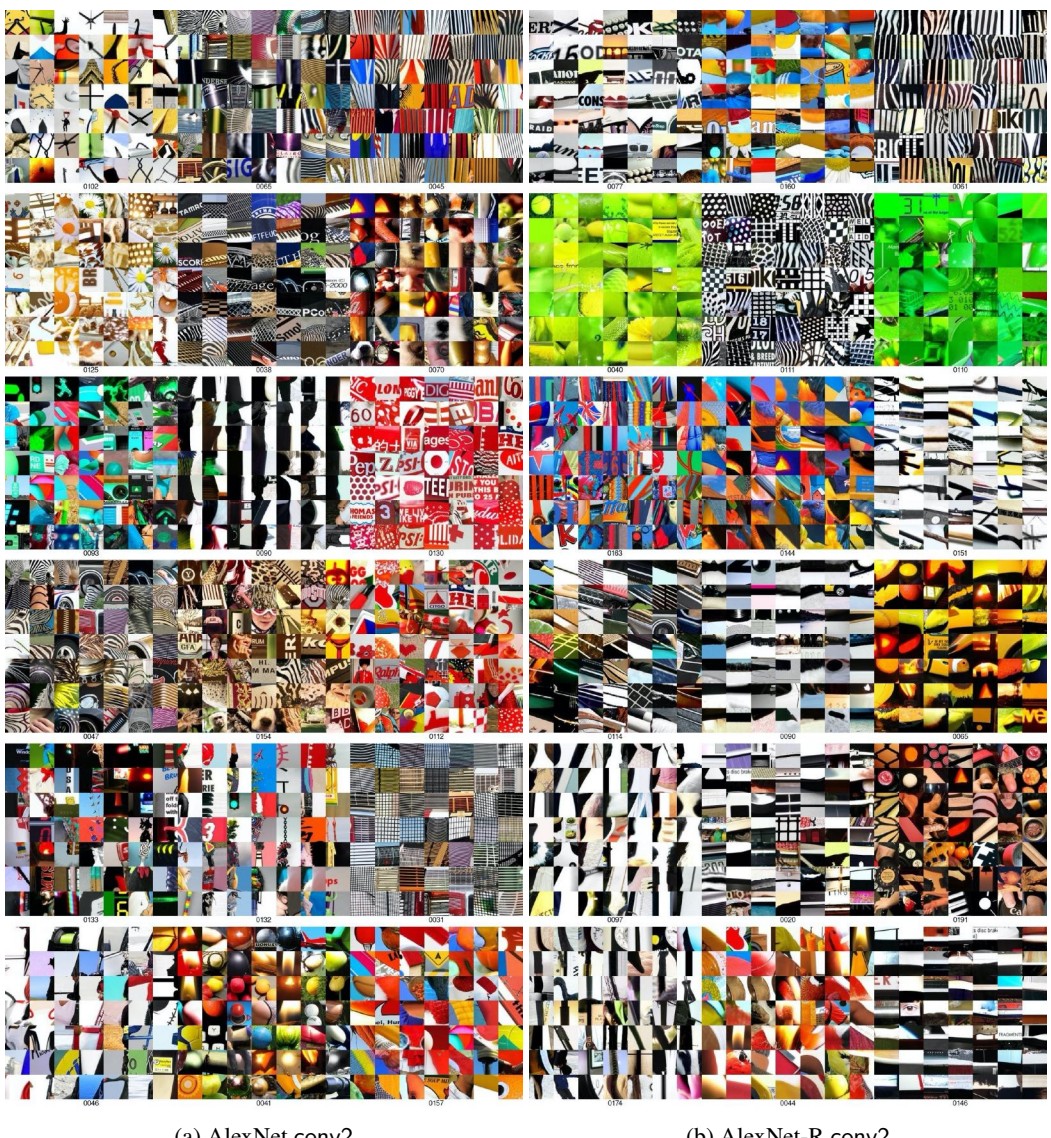

(a) AlexNet conv2                    (b) AlexNet-R conv2

Figure A16: Each $7 \times 7$ grid shows the top-49 training-set images that highest activate the center unit in a channel for AlexNet (a) and AlexNet-R (b). A set of 20 channels were randomly sampled from the 192 channels in conv2 of the networks. The top-49 of AlexNet-R comprises of more just green color, edge, and chequered pattern detector channels. For a complete set of 192 channels, refer to https://drive.google.com/drive/folders/1eNmkxYT1nSBO-EWoeCkFDXAH60Rf8VrU?usp=sharing.

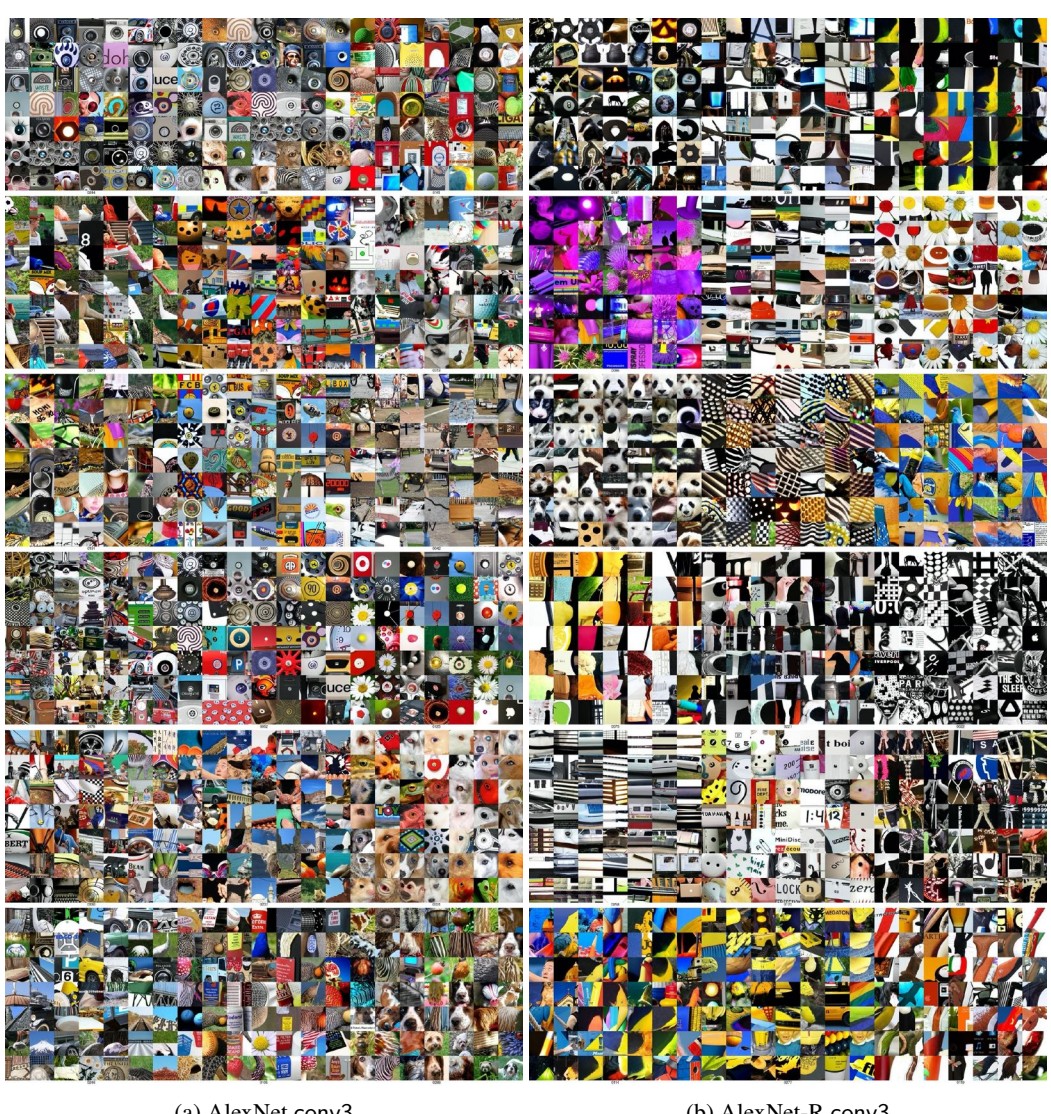

(a) AlexNet conv3           (b) AlexNet-R conv3

Figure A17: Each $7 \times 7$ grid shows the top-49 training-set images that highest activate the center unit in a channel for AlexNet (a) and AlexNet-R (b). A set of 20 channels were randomly sampled from the 384 channels in conv3 of the networks. The top-49 of AlexNet-R comprises of more simple edge and dotted pattern detector channels. For a complete set of 384 channels, refer to https://drive.google.com/drive/folders/1eNmkxYT1nSBO-EWoeCkFDXAH6ORf8VrU?usp=sharing.

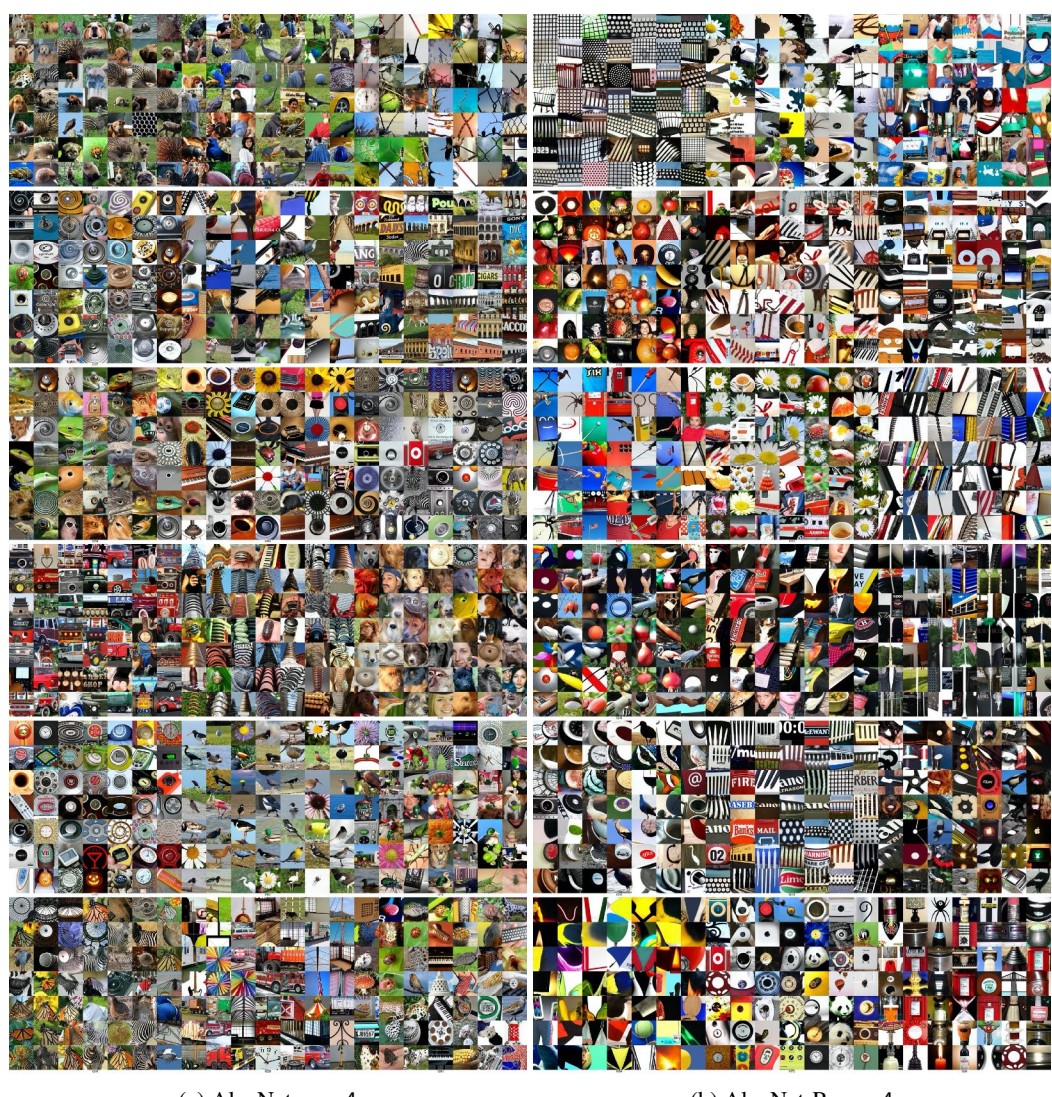

(a) AlexNet conv4                                        (b) AlexNet-R conv4

Figure A18: Each $7 \times 7$ grid shows the top-49 training-set images that highest activate the center unit in a channel for AlexNet (a) and AlexNet-R (b). A set of 20 channels were randomly sampled from the 256 channels in conv4 of the networks. The top-49 of AlexNet-R comprises of more simple texture detector channels whereas we observe face detector channels in AlexNet (a). For a complete set of 256 channels, refer to https://drive.google.com/drive/folders/1eNmkxYT1nSBO-EWoeCkFDXAH60Rf8VrU?usp=sharing.

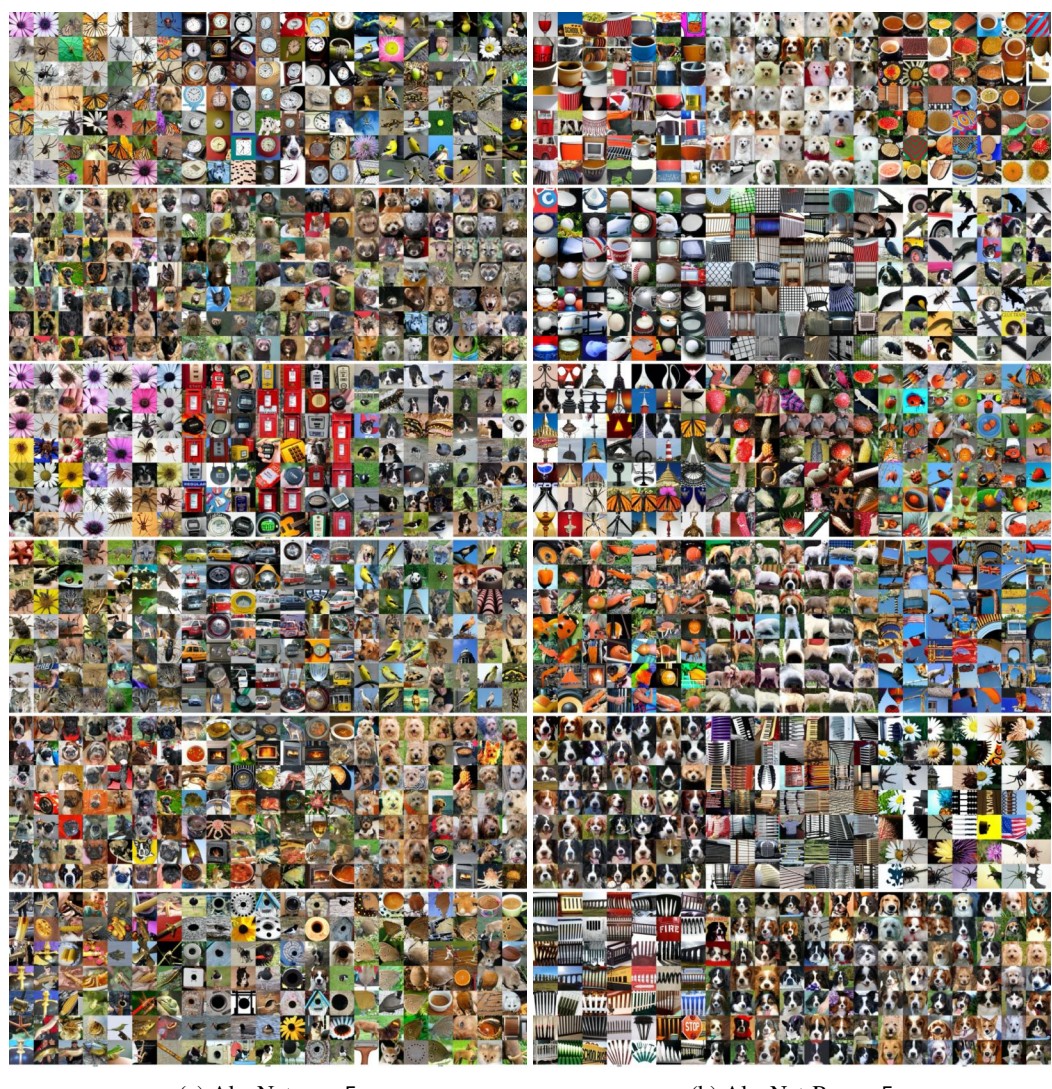

    (a) AlexNet conv5                                       (b) AlexNet-R conv5

Figure A19: Each $7 \times 7$ grid shows the top-49 training-set images that highest activate the center unit in a channel for AlexNet (a) and AlexNet-R (b). A set of 20 channels were randomly sampled from the 256 channels in conv5 of the networks. The top-49 of AlexNet-R comprises of less diverse object detector channels whereas some AlexNet channels activate for both daisy and spider. For a complete set of 256 channels refer to `https://drive.google.com/drive/folders/1eNmkxYT1nSBO-EWoeCkFDXAH60Rf8VrU?usp=sharing`.

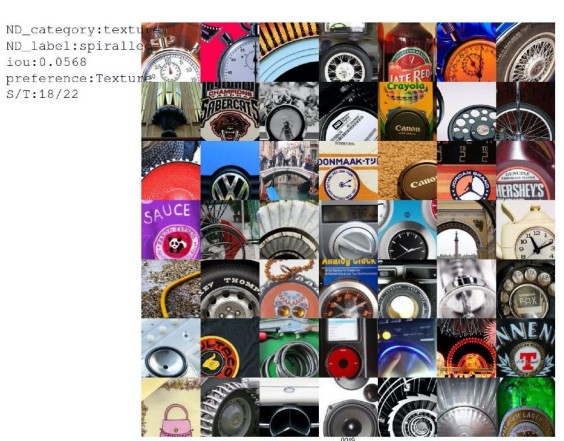

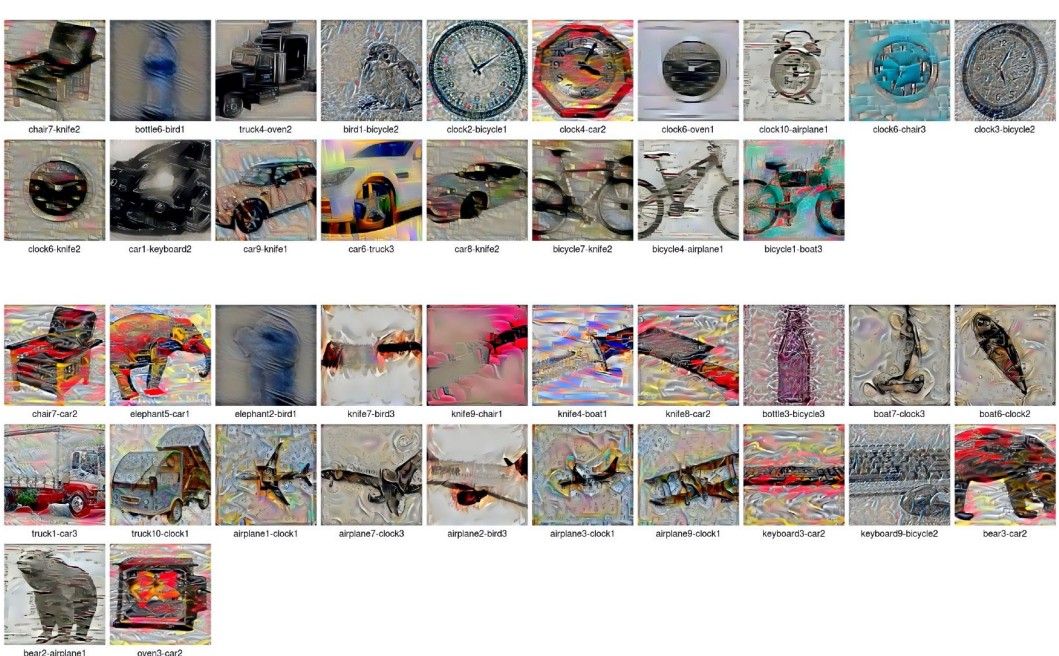

Figure A20: AlexNet conv4$_{19}$ with Shape and Texture scores of 18 and 22, respectively. It has a NetDissect label of spiralled (IoU: 0.0568) under texture category. Although this neuron is in NetDissect texture category, the misclassified images suggest that this neuron helps in both shape- and texture-based recognition. **Top:** Top-49 images that highest-activated this channel. **Middle:** Mis-classified images in shape category (18 images). **Bottom:** Mis-classified images in texture category (22 images).

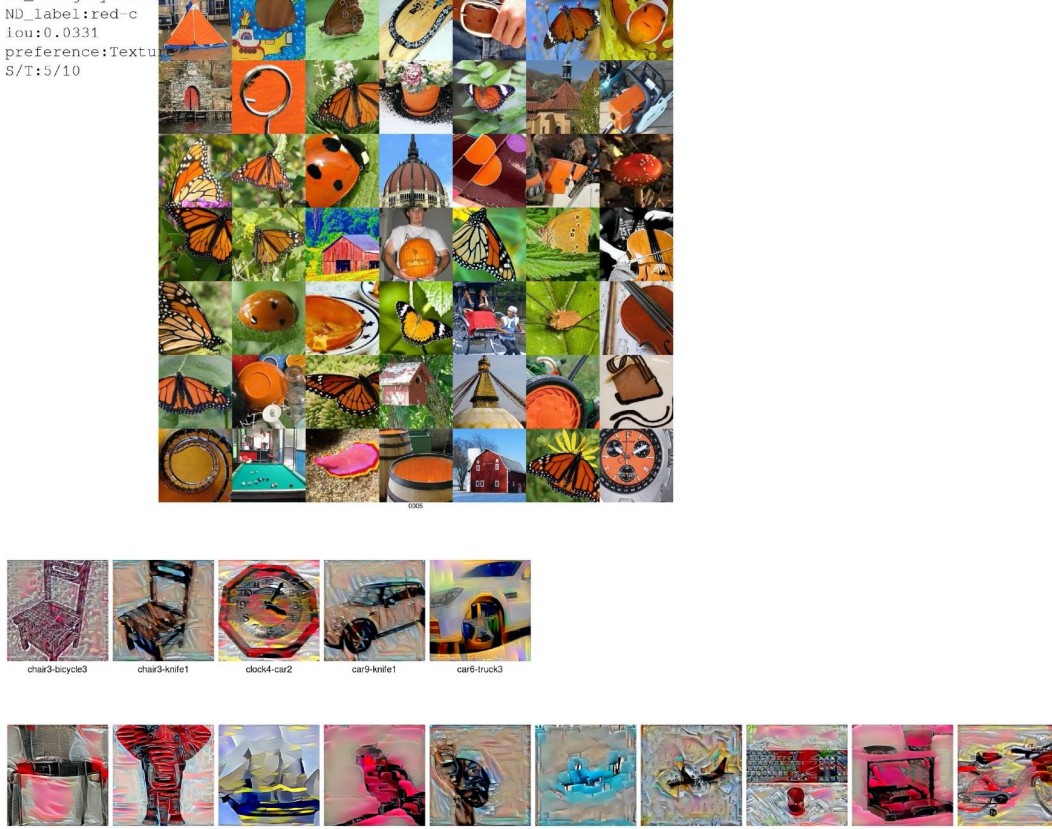

Figure A21: AlexNet conv5$_5$ with Shape and Texture scores of 5 and 10, respectively. It has a NetDissect label of red (IoU: 0.0331) under color category. But this neuron also detects circular shapes. **Top:** Top-49 images that highest-activated this channel . **Middle:** Mis-classified images in shape category. **Bottom:** Mis-classified images in texture category.

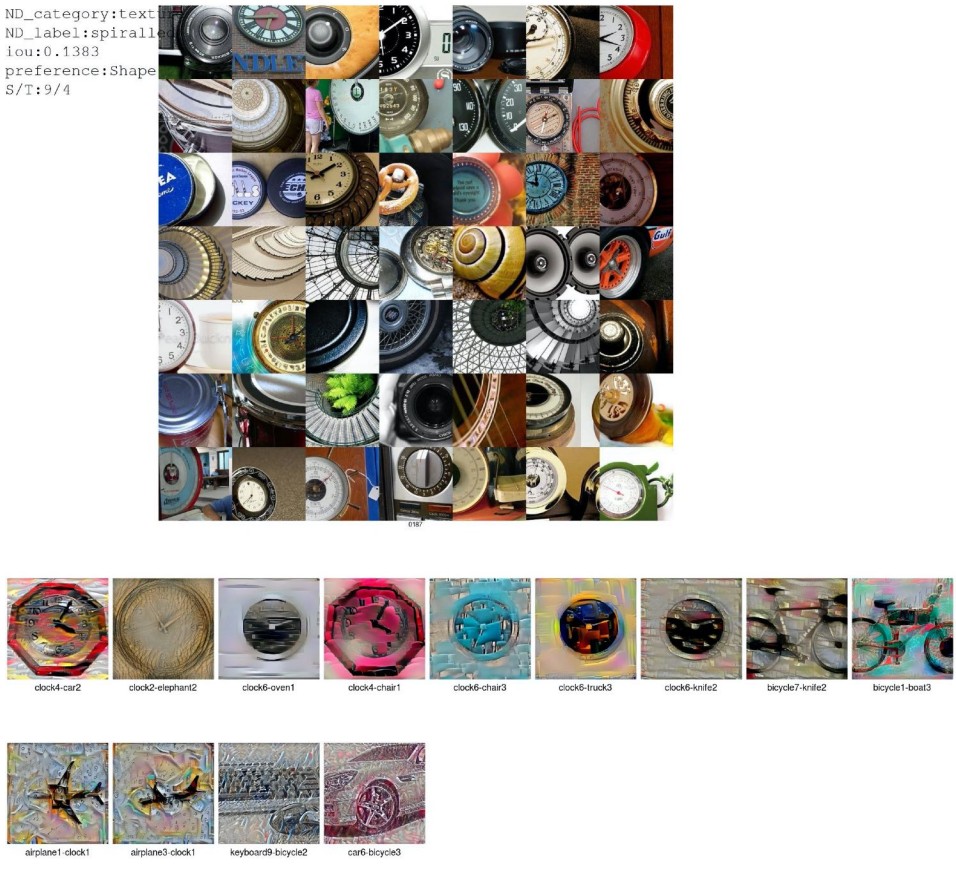

Figure A22: AlexNet conv5$_{187}$ with Shape and Texture scores of 9 and 4, respectively. It has a NetDissect label of spiralled (IoU: 0.1383) under texture category. A shape-biased neuron that detects objects of circular shapes e.g. clock and wheels. **Top:** Top-49 images that highest-activated this channel. **Middle:** Mis-classified images in shape category. **Bottom:** Mis-classified images in texture category.

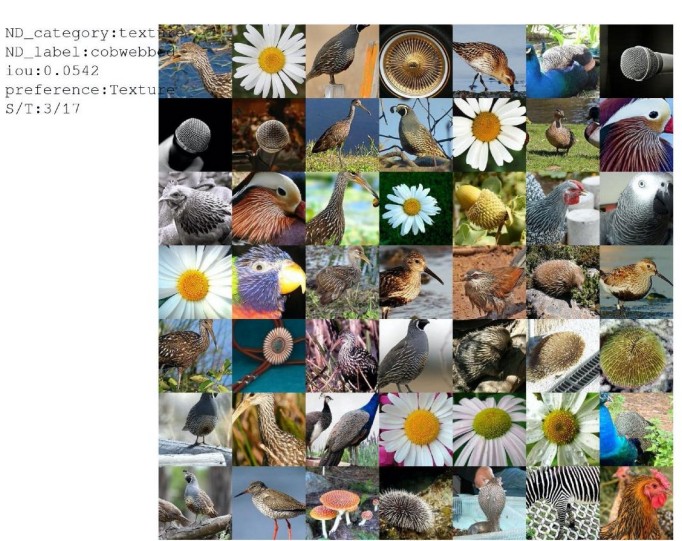

ND_category:texture
ND_label:cobwebbed
iou:0.0542
preference:Texture
S/T:3/17

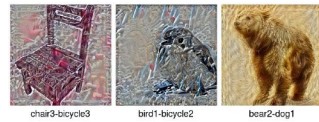

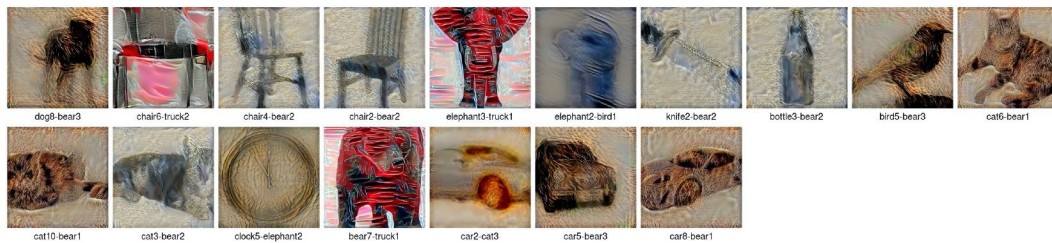

Figure A23: AlexNet conv5$_{221}$ with Shape and Texture scores of 3 and 17, respectively. It has a NetDissect label of cobwebbed (IoU: 0.0542) under texture category. This is a heavily texture-biased neuron that helps networks detect animals by their fur textures. **Top:** Top-49 images that highest-activated this channel. **Middle:** Mis-classified images in shape category. **Bottom:** Mis-classified images in texture category.

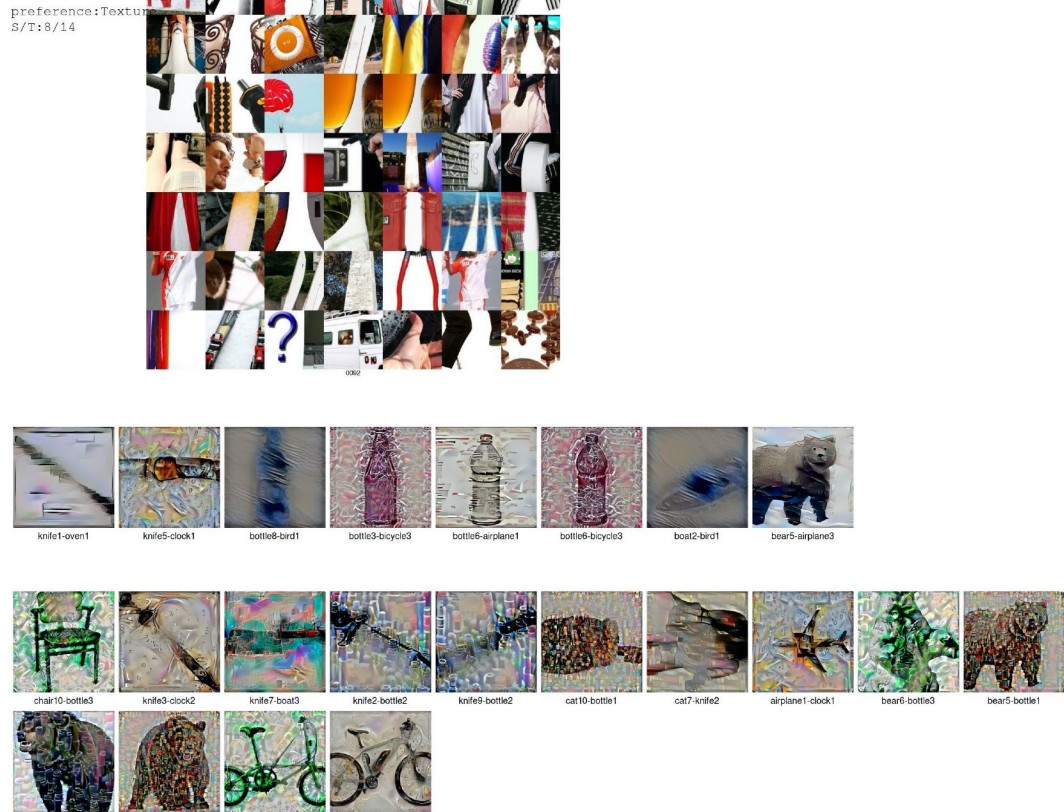

Figure A24: AlexNet-R conv4$_{92}$ with Shape and Texture scores of 8 and 14, respectively. It has a NetDissect label of white (IoU: 0.0143) under color category. Combining the top-49 images with the mis-classified shape images, this neuron seems to detect shapes of bottles. **Top:** Top-49 images that highest-activated this channel. **Middle:** Mis-classified images in shape category. **Bottom:** Mis-classified images in texture category.

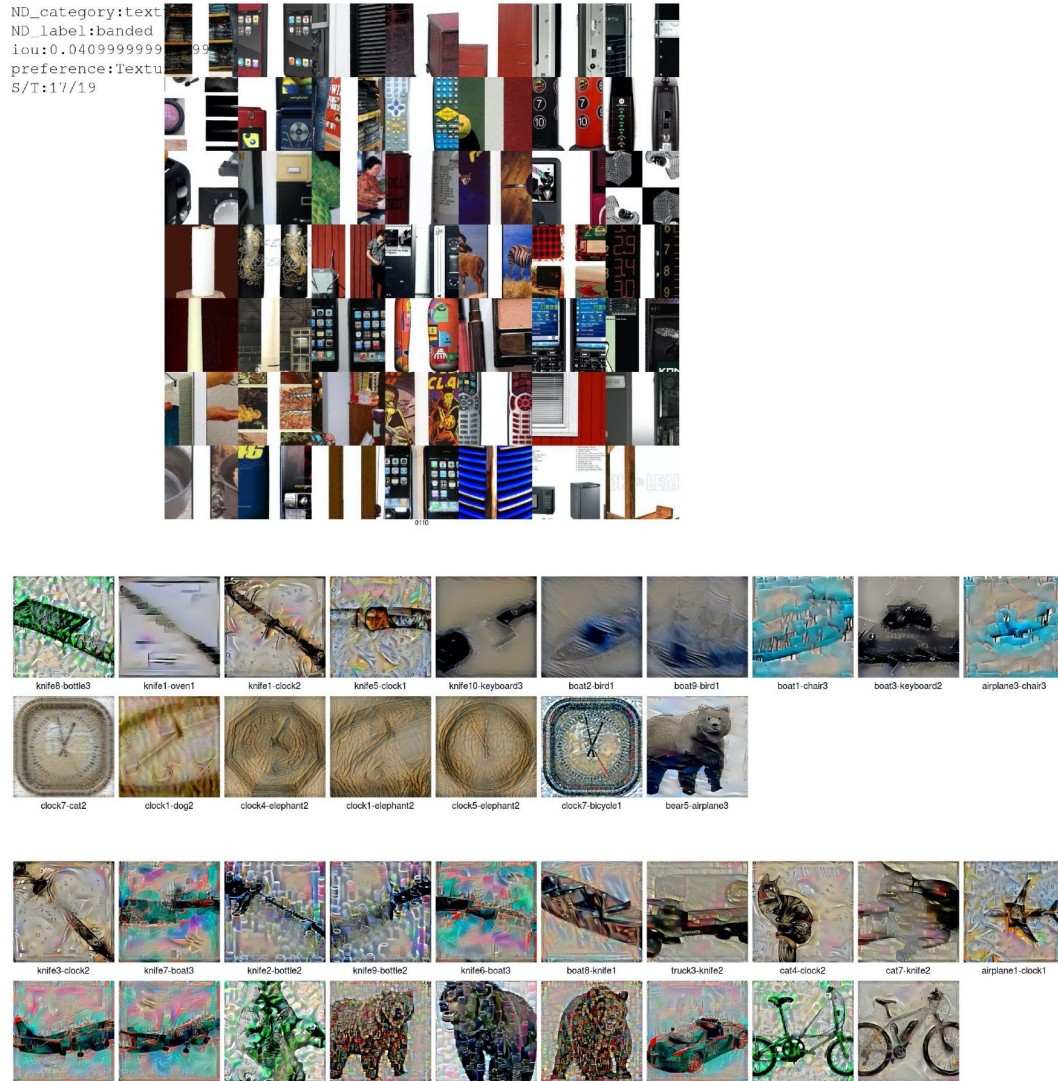

Figure A25: AlexNet-R conv5$_{110}$ with Shape and Texture scores of 17 and 19, respectively. It has a NetDissect label of banded (IoU: 0.0409) under texture category. This neuron has almost equal Shape and Texture scores and is useful in detecting both the shape and textures of knives and bottles at the same time. **Top:** Top-49 images that highest-activated this channel. **Middle:** Mis-classified images in shape category. **Bottom:** Mis-classified images in texture category.

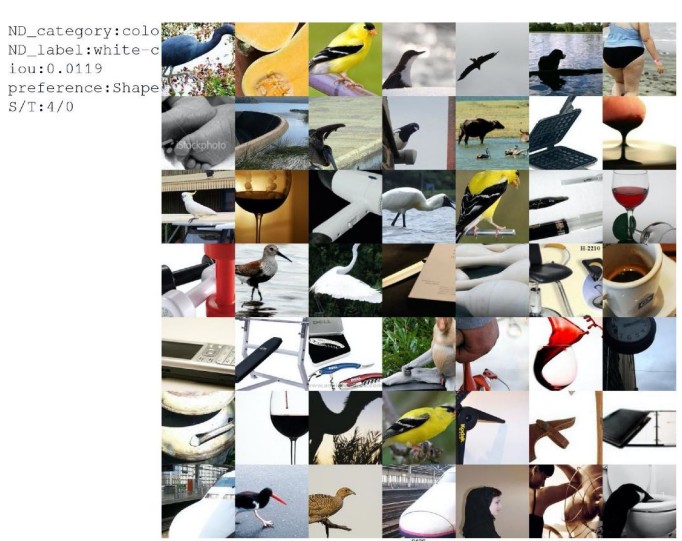

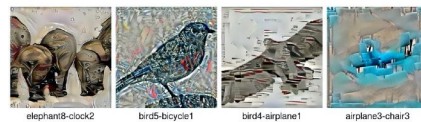

Figure A26: AlexNet-R conv5$_{136}$ with Shape and Texture scores of 4 and 0, respectively. It has a NetDissect label of white (IoU: 0.0119) under color category. Interestingly, this neuron is heavily shape-biased. **Top:** Top-49 images that highest-activated this channel. **Middle:** Mis-classified images in shape category. **Bottom:** Mis-classified images in texture category.

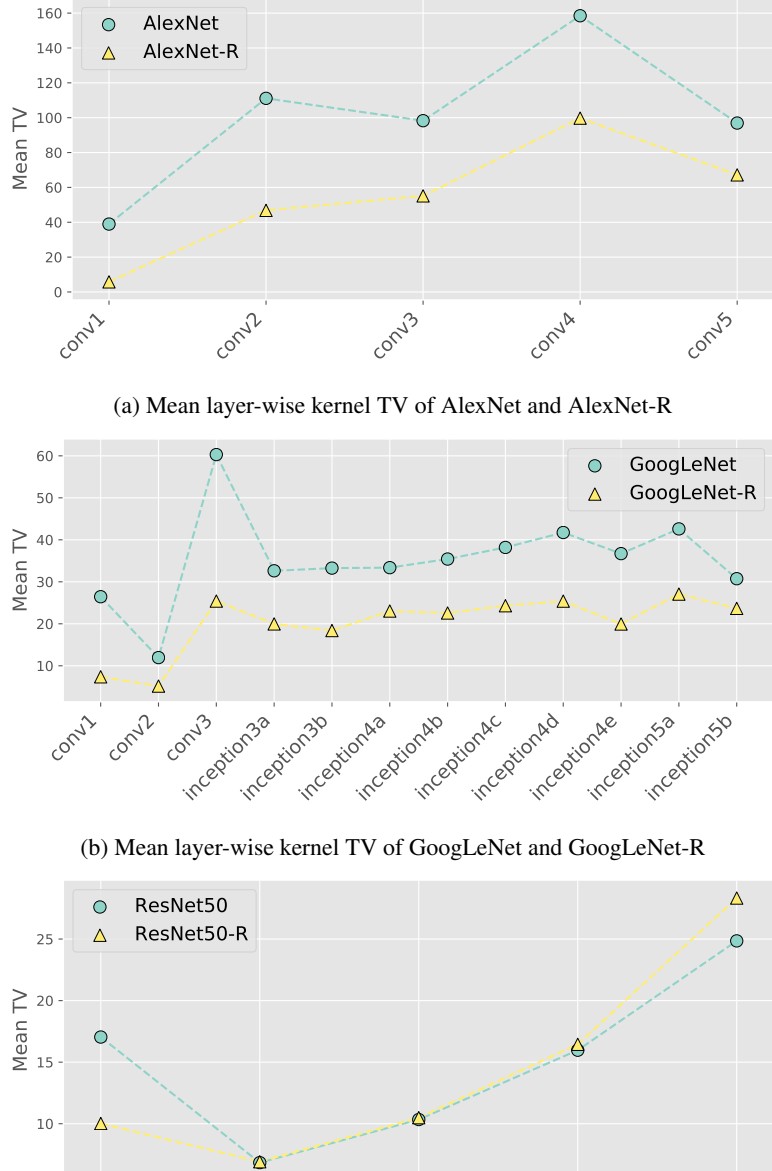

(a) Mean layer-wise kernel TV of AlexNet and AlexNet-R

(b) Mean layer-wise kernel TV of GoogLeNet and GoogLeNet-R

(c) Mean layer-wise kernel TV of ResNet and ResNet-R

Figure A27: In early convolutional layers, the kernels is much smoother in robust network. i.e. (a)&(b) the kernel is smoother in first few conv layers. (c) is a bit special since the layer∗ is residual blocks, but we can still see that conv1 in robust network has much smoother kernel compared to its' counter part. *Note:We compute the mean TV of all* conv *layers inside each residual(inception) blocks.*

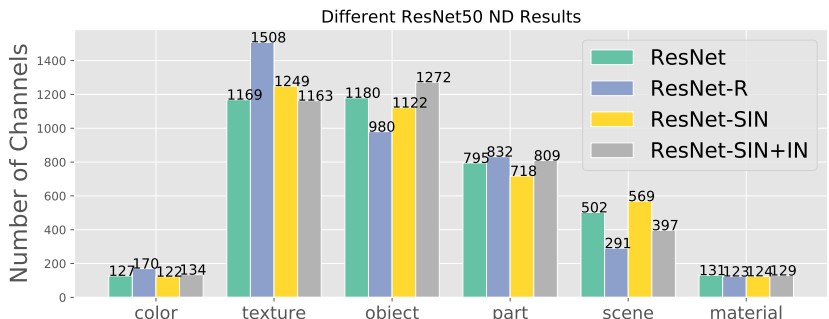

Figure A28: NetDissect results of ResNet50 trained differently. It is obvious that ResNet-R is very special compared to all other variance. An interesting trend is that shape biased model (ResNet-R, ResNet-SIN) tends to have more texture detectors and fewer object detectors, wise verser.

