# OpenReview forum: "The shape and simplicity biases of adversarially robust ImageNet-trained CNNs"
_ICLR.cc/2021/Conference — Reject_

### Official Review · AnonReviewer4 · 2020-10-25
**Fairly thorough experimental study, few new insights or consequences**

**Rating:** 6
**Confidence:** 4

**Review:**

Summary:
The submission concerns an experimental study of the behavior of networks trained with and without adversarial robustness criteria (using Madry et al., 2017). Given a set of such trained networks, a detailed look at the behavior and properties of adversarially robust networks and their non-robust counterparts is taken. This includes evaluation on the cue-conflict dataset (Geirhos et al.), and on scrambled, or texture-less (silhouette) variants of ImageNet images. Furthermore, visualizations of filter banks are analyzed and compared, as well as an analysis on the neuron level is carried out, using the NetDissect framework by Bau et al.
Insights include that adversarially trained networks are more shape-biased (and reliant) than their counterparts which are known to be texture-biased. Furthermore, three simplicity biases are found that result in smoother filters, increased focus on low-level cues, and decreased diversity of inputs detected by neurons of adversarially trained networks.

Review:
The authors have conducted a fairly thorough experimental study w.r.t. the level of detail considered; insights range from experiment to neuron level. The main limitations of the study are set by the types of networks considered (2012-2015 era networks only) and the types of adversarial training considered (only the work by Madry et al.). With regards to the latter, incorporation of newer work (e.g. (1), (2), or (3)) on adversarial training could have made the argument stronger, but this may simply have been subject due to scope and time limitations.
Despite these limitations, the scope of evaluation is sufficient, and the level of supplied visualizations is exemplary. The work lacks any theoretical insight, but there is potential value in the type of conducted detailed empirical study. A large part of the results seems to agree with prior work (cf. mentions of “consistent with” and similar), such that part of the value of this study is additional confirmation of what prior work may have found or otherwise hypothesized.
What I am missing a bit, however, is a discussion on what the mentioned findings may mean for future developments of adversarial training, adversarial attack design, or other mitigations of adversarial attacks. The discussion of possible consequences is limited to the last paragraph, which is meant to discuss future work.

Clarity of writing is generally good; however, I was not really a fan of the chosen nomenclature and the abundant definition of abbreviations which may clash with own notions of their meaning. For example, is it necessary to define “S-networks” and “R-networks”, where R-networks happen to be *s*hape-biased and S-networks *t*exture-biased? My preference would have been to simply spell out “adversarially trained” (“-adv”) vs. “not adversarially trained”. This is just a personal opinion and does not affect the rating. Similarly with the datasets; “ImageNet-C” is mentioned on page 3, but does not seem to be defined beyond “modified ImageNet images” and then shown in Figure 1.

Overall, I believe the scope of experiments and evaluations goes beyond workshop-level work. But originality and significance remain limited, as mentioned above.

—-
(1) Shafahi et al., “Adversarial Training for Free!”, NeurIPS 2019.
(2) Cohen et al., “Certified Adversarial Robustness via Randomized Smoothing”, ICML 2019.
(3) Xie et al., “Smooth Adversarial Training”, arXiv preprint, 2006.14536

---

> ### Author Response · Authors · 2020-11-25
> **Replies to R4**
>
> Thank you very much for your positive feedback and constructive comments!
> Please find our inline responses below:
>
> >**The main limitations of the study are set by the types of networks considered (2012-2015 era networks only)**
>
> A reason we used AlexNet, GoogLeNet, and ResNet-50 is that these networks have been the subject of studies in neuroscience [3] and interpretability work in ML/CV e.g. [1][2][4]. Studying these networks naturally extends the insights previously discovered in the literature.
> AlexNet has become now a major hypothesis for how human brains may work [3].
>
>
> - [1] Synthesizing the preferred inputs for neurons in neural networks via deep generator networks. Nguyen et al. NIPS 2016
> - [2] Network dissection: Quantifying interpretability of deep visual representations. Bau et al. CVPR 2017
> - [3] https://www.nature.com/articles/s41598-017-13756-8
> - [4] ImageNet-trained CNNs are biased towards texture; increasing shape bias improves accuracy and robustness. Geirhos et al. ICLR 2019
>
>
>
> >**What I am missing a bit, however, is a discussion on what the mentioned findings may mean for future developments of adversarial training, adversarial attack design, or other mitigations of adversarial attacks. The discussion of possible consequences is limited to the last paragraph, which is meant to discuss future work.
>
> We have updated the Discussion section to elaborate more on the utility of the findings in our paper.
>
> Our main contributions are the novel insights into the inner-workings of adversarially robust models and such interesting findings can be in turn harnessed to develop useful future algorithms. For example:
>
> - We found that R networks heavily rely on shape cues. One may fuse an S network and an R network (two channels, one uses texture and one uses shape) into a single, more robust, interpretable ML model.
> That is, such a model may (1) have better generalization on OOD data than S or R network alone and (2) enable an explanation to users on what features a network uses to label a given image.
>
> - We found adversarial training to induce major, consistent internal neural changes to a network. Such findings, e.g. smoother filters (Sec. 3.3.1), neurons detecting lower-level and fewer features (Sec. 3.3.2), can be formulated into a regularizer that can be incorporated into the training of a vanilla network performance (instead of adversarial training which requires ~2x longer training time due to the computation of adversarial examples).
>
> - Our study on how individual hidden neurons contribute to the R network shape preference (Sec. 3.4)  revealed that texture-detector units are equally important to the texture-based and shape-based recognition. This is in contrast to a common hypothesis that texture detectors should be exclusively only useful to texture-biased recognition. This surprising finding suggests that the categories of stimuli in the well-known Network Dissection (Bau et al. 2017; 2019) need to be re-labeled and also extended with low-frequency patterns e.g. single lines or silhouettes in order to more accurately quantify hidden representations.

---

### Official Review · AnonReviewer1 · 2020-10-26
**Empirical studies on the adversarial trained CNNs and shape-based representation**

**Rating:** 6
**Confidence:** 4

**Review:**

This paper takes a step further to understand the relationships between the adversarial trained CNNs (R-CNNs) and shape-based representation, and delve deeper into the R-CNNs via studying the hidden units. First, it justifies that the R-CNNs prefer shape cues based on random-shuffled, Stylized-ImageNet, and silhouette experiments. Then, it tests R-CNNs on ImageNet-C to show the less connection between the shape-biased and the robustness against common corruptions. Finally, it studies the hidden unit via qualitative tools including NetDissect.

The studied direction is important for both representation learning and robustness communities. The methodology of this paper and all experiments are technically sound. Using Network Dissection is a good point to study here. Testing on three benchmarks, and evaluate three different network architectures can help to verify the conclusions are general. The results are sufficient and would support their arguments, though more analysis would add support to the claims. Overall the paper is easy to follow and the reviewer thinks the experiments are easy to replicate.

---------------
Below are some concerns and suggestions:

The novelty is slightly limited. Many understanding is complementary to the findings in the literature, such as [1] that adversarially trained models are shape-biased, [2] that there're nor significant correlations between shape-biased and robustness against common corruptions, and [3] that low-frequency help the generalization. Many experimental methods are the same as previous work but only perform on adversarially trained models, like the scrambled (Sec. 3.2.1) is the same as the random shuffled in [4],  test on Stylized-ImageNet[2], ImageNet-C, and etc. Also, there're no new techniques proposed. Basically, all the used techniques are from literature, like Network Dissection, and etc.


According to Table 3, compare to the Shape-Less column, the difference in the Texture-less column is not that significant. The reviewer wonder if the author would perform extra experiments to verify the R models significantly more prefer shape information, such as test on edge maps. The edge map is easy to drive from the silhouette and may be more suitable to be tested here. The value of the edge map and silhouette may be set to binary or greyscale [0, 255].

Currently, only qualitative results are provided to justify the R models would contain smoother filters than the standard ones. The reviewer wonders if any quantitative criterion can be designed and be reported thereon.

In the blocking pixel-wise noise section (Page 6), the author claims that R models are more robust against Gaussian additive noise based on Figure 3. However, the ImageNet-C contains such distorted images, and S models outperform than R models. The reviewer wonders if the author can test on Gaussian additive noise distorted images and report the results to further justify the claim.

---------------------------------
To sum up, the reviewer would vote 5 currently -- due to its limited novelty but some understanding of the representation of adversarially trained models are proposed.

[1]Interpreting Adversarially Trained Convolutional Neural Network

[2]Imagenet-trained Cnns are Biased towards Texture; Increasing Shape Bias improves Accuracy and Robustness

[3]High-frequency Component Helps Explain the Generalization of Convolutional Neural Networks

[4]Defective Convolutional Layers Learn Robust CNNs


-------------------after rebuttal-----------------------

I thank the authors for their rebuttal. Since the authors reply near the discussion phase end, I cannot ask follow-up questions.

The answers partially address my concerns and thus I would raise my score to 6.

For novelty , the author answers three points. For the first point, fuse two channels of information are not so convincing on the novelty aspects. Also, there needs an extra cost to collect process the images. For the second point, how would you formulate a regularize?

---

> ### Author Response · Authors · 2020-11-25
> **Replies to R1**
>
> Thank you very much for your positive feedback and constructive comments!
> Please find our inline responses below:
>
> >**novelty is slightly limited**
>
> Our main contributions are the novel insights into the inner-workings of adversarially robust models and such interesting findings can be in turn harnessed to develop useful, future algorithms. For example:
>
> - We found that R networks heavily rely on shape cues. One may fuse an S network and a R network (two channels, one uses texture and one uses shape) into a single, more robust, interpretable ML model.
> That is, such model may (1) have better generalization on OOD data than S or R network alone and (2) enable an explanation to users on what features a network uses to label a given image.
>
> - We found adversarial training to induce major, consistent internal neural changes to a network. Such findings, e.g. smoother filters (Sec. 3.3.1), neurons detecting lower-level and fewer features (Sec. 3.3.2), can be formulated into a regularizer that can be incorporated into the training of a vanilla network performance (instead of adversarial training which requires ~2x longer training time due to the computation of adversarial examples).
>
> - Our study on how individual hidden neurons contribute to the R network shape preference (Sec. 3.4)  revealed that texture-detector units are equally important to the texture-based and shape-based recognition. This is in contrast to a common hypothesis that texture detectors should be exclusively only useful to texture-biased recognition. This surprising finding suggests that the categories of stimuli in the well-known Network Dissection (Bau et al. 2017; 2019) need to be re-labeled and also extended with low-frequency patterns e.g. single lines or silhouettes in order to more accurately quantify hidden representations.
>
> We have added these points into the Discussion and Future work section.
>
>
> >**The reviewer wonder if the author would perform extra experiments to verify the R models significantly more prefer shape information, such as test on edge maps.**
>
> Thank you for the suggestion! We did run this test.
>
> We followed generated binary contour images (see Fig. 1f) and found that R model performance is 1.6 x higher than S model performance (see updated Table 3; contour column).
> Separately, we also ran another experiment on edge images produced by canny-edge; however, R models only slightly outperformed S models (higher by a few percent).
>
>
> >**Currently, only qualitative results are provided to justify the R models would contain smoother filters than the standard ones. The reviewer wonders if any quantitative criterion can be designed and be reported thereon.**
>
> Thanks! In light of your suggestions, we compare directly the smoothness of the convolutional filters (in Total Variation). The conv filters in R networks are substantially smoother than those in S networks (see Sec. 3.3.1). For example, AlexNet-R is 1.7x smoother than AlexNet (63.59 vs. 110). For all three architectures, R networks are smoother in early layers (see Fig. A27).

---

### Official Review · AnonReviewer2 · 2020-10-30
**Review of The shape and simplicity biases of adversarially robust ImageNet-trained CNNs**

**Rating:** 5
**Confidence:** 4

**Review:**

In this paper, the authors show that adversarially robust versions of three popular CNN architectures trained for image classification on ImageNet rely on shape rather than on textures to perform recognition. They also show that adversarially robust networks do not outperform non-robust networks on corrupted data. Finally, they perform some analysis to determine whether intermediate features are more related to shape or texture, finding that these representations to intertwine both types of information.

**Quality:** The paper is well written and the experiments are very interesting. However, the contributions may be somewhat incremental (see below).

**Clarity:** The paper is written very clearly.

**Significance and originality:** The paper is interesting, but the novelty is a bit limited. I am torn about this, because I do think that the experiments are interesting and well explained, and verifying results on different datasets (especially very large datasets like ImageNet) is important.  However,  given that the shape bias of adversarially robust networks was already known previously (as pointed out by the authors) and that there is no methodological contribution, I think that the contributions are somewhat incremental.

**Pros:** Well written, interesting experiments.

**Cons:** Incremental contribution.

After reading the author feedback, I would like to thank the authors and I agree with them that it is critical to test hypotheses on large-scale datasets. However, I still think that the contribution is marginally below the acceptance threshold.

---

> ### Author Response · Authors · 2020-11-25
> **Replies to R2**
>
> Thank you very much for your positive feedback and constructive comments, which made our paper even stronger!
> Please find our inline responses below:
>
> >**The paper is interesting, but the novelty is a bit limited.**
>
> Let us clarify the novelty and therefore the contribution:
>
> - Compared to the published prior work, we are the first to study the shape and texture bias of ImageNet models. Our work is a natural extension to the texture-bias question, which was first studied on *ImageNet* models in [1].
> Zhang et al. 2019 performed similar tests but on *smaller-scaled* datasets (CIFAR-10, Caltech256), which are not guaranteed at all to generalize to ImageNet (1000-class, 256x256 images) both due to large differences in network-architecture scales, number of classes, and image resolution.
>
> - We are the first to quantitatively characterize the hidden representations of adversarially robust networks, here using the Network Dissection framework (Sec. 3.3) and cue-conflict (Sec. 3.4). Previous work e.g. Santukar et al. 2019 & Engstrom et la. 2019 only qualitatively compare standard vs. robust features via image synthesis.
>
> - Human perception is much more robust to adversarial examples and common out-of-distribution images; however, machine vision is notoriously brittle. Geirhos et al. [1] results suggested that a reason for human vision’s robustness is that humans focus heavily on shape cues instead of texture. Here, our study naturally fills in the gap by answering: Are adversarially robust networks also shape-biased? If so, what internal mechanisms inside adversarially-robust networks made them more robust and shape-biased?
> On ImageNet, we are the first to address this question.
>
> [1] The shape and simplicity biases of adversarially robust ImageNet-trained CNNs. Geirhos et al. ICLR 2019.
>
>
> >**that there is no methodological contribution, I think that the contributions are somewhat incremental.**
>
> Machine learning field has witnessed many methods working on MNIST but do not generalize any to larger-scaled datasets. Similarly, we strongly believe this type of interpretability work needs to be done on ImageNet and our such first contribution is valuable to the community.
> We agree that we did not propose a new method; however, our work is a typical neuroscience type of work that harnesses existing tools to characterize existing models.

---

### Official Review · AnonReviewer5 · 2020-11-04
**Review of "The shape and simplicity biases of adversarially robust ImageNet-trained CNNs"**

**Rating:** 3
**Confidence:** 4

**Review:**

The authors show that adversarially trained models have higher shape and simplicity biases compared to their vanilla counterparts.
The paper is well written and easy to follow. I think I understood most parts of the paper. The authors were explicit in stating which methods they used which is good.
My major concerns are:
1.	Lacking novelty/ comparison to previous work: The authors did not compare to previous work properly. In my general comments, I suggest adding and discussing 11 additional publications. One of their major findings that adversarially trained models are invariant to additive high frequency noise has been shown and discussed in [8] which they did not cite. Other cited papers are cited in a wrong way, e.g. Geirhos et al 2018b did not perform experiments on adversarial examples; there are other examples in the General comments section below.
2.	Not relevant/ not surprising results: I am not sure how useful and interesting the finding is that adversarially robust models are more shape biased. It seems intuitive that given that adversarial perturbations are high-frequency, becoming invariant to them should make a model more attentive to low-frequency features such as shape.
3.	Lacking discussion/ analysis of the results: In the first experiments section, the authors found that adversarially robust models are shape biased. With the NetDissect analysis, they showed that there are more filters in adversarially robust models compared to vanilla models that detect textures. These two results seem contradictory to me and should be discussed. If they are not contradictory, this should be explained.
4.	The number of PGD steps used for the adversarial attack is 7 which is very low. I am not sure whether the results in this paper are meaningful in the sense that the models might not be adversarially robust at all if a higher number of PGD steps/random restarts were used. The authors should test bigger numbers of steps and make the step size dependent on the number of steps.


General comments:

Page 2 “In contrast, there was also evidence that classifiers trained on one type of adversarial example often do not generalize well to other image types [Geirhos et al, 2018b, Nguyen et al., 2015]”. What is image types here? Both references do not provide any support for the claim made in this sentence. In Gheiros et al, the authors train on different types of image corruptions such as additive uniform noise or using high/loss pass filters. They do not show results for adversarial examples. Nguyen et al was published two years before adversarial training was proposed by Madry et al. There are other references that would be good to cite here though: [1] show that increasing robustness against adversarial attacks does not increase robustness against translations and rotations. [2] show that adversarial robustness does not transfer easily between attack classes. [3] also show that there is a performance trade-off between robustness types.

Page 3: “Adversarial examples were generated using Projected Gradient Descent (PGD) (Madry et al., 2017) with an L2 norm constraint of 3, a step size of 0.5, and 7 PGD-attack steps.” A number of 7 PGD steps seems very very low. A more sensible number might be e.g. 1000 [4] or at least 200-400 steps. In [5], the authors urge to check the convergence of the attack and that white-box attacks generally converge in under 100-1000 steps. Additionally, Carlini et al [5] write that “the number of iterations necessary is generally inversely proportional to step size and proportional to distortion allowed.” -> Here, a step-size of 0.5 is chosen without providing any reasoning and the choice seems arbitrary.

Table 1: On the torchvision website, the top1 accuracies for the S models are given as: AlexNet: 56.55%, GoogleNet: 69.78%, ResNet50: 76.15%. These numbers differ from what the authors show in Table 1. Working with the pretrained torchvision models myself every day, I know that a ResNet50 from torchvision will get a top1 accuracy of 76.13% (same to what is stated on the used “robustness” package repo of Engstrom et al). Where does the difference to the numbers in Table 1 come from?

Page 4: “R models show no generalization boost on ImageNet-C i.e. they performed on-par or worse than the S counterparts (Table 3c). This is consistent with the findings in Table 4 & 5 in Geirhos et al. (2018a) that a stronger shape bias does not necessarily imply better generalizability.” -> This is completely opposite to what we see in Table 5 in Geirhos et al. 2018a. There, the authors attribute the higher robustness to common corruptions to the higher shape bias of their SIN model: “Again, none of these corruption types were explicitly part of the training data, reinforcing that incorporating SIN in the training regime improves model robustness in a very general way.” Due to the contradictory results, it rather seems that shape bias and corruption robustness might be more disconnected than previously thought. Also, the result that adversarially trained networks perform badly on ImageNet-C has been observed in [6] which might be good to cite here.

Page 4: “However, this bias is orthogonal to the performance when only either texture or shape cues are present.” I don’t understand why it would be orthogonal? I would expect a model with a strong shape bias to care less if I remove texture cues from an image compared to a model with a strong texture bias. What exactly is new in this test compared to the cue conflict task?

Page 6: “To confirm this hypothesis”, please write “to test this hypothesis”

Page 6: “That is, the smooth filters in R models indeed can filter out pixel-wise Gaussian noise despite that R models were not explicitly trained on this image type!” The finding here seems very related to [7] and [8]. In [7], the authors show that adversarial training improves robustness to Gaussian noise and in [8], the authors study the Fourier properties of common corruptions and show that adversarial training improves accuracy for corruptions in the higher frequencies.

Page 6: “Our result suggests that most of the de-noising effects take place at lower layers (which contain generic features) instead of higher layers.” I do not understand how this conclusion was made given the results, please explain.

Page 6: “We found a consistent trend—adversarial training resulted in substantially more filters that detect colors and textures (i.e. in R models) in exchange for fewer object and part detectors”. This result seems surprising and not consistent with the finding that adversarially robust models are more shape biased. If shape bias is indeed correlated to the filter responses (which is not necessarily clear), I would expect filters of a shape-biased model to rather detect objects and parts instead of textures and colors. Please discuss this. It would also be nice to test the SIN trained model from Geirhos et al 2018a, since it has a high shape bias.

Page 7: “As R hidden units fire for fewer concepts, i.e. significantly fewer inputs, the space for adversarial inputs to cause R models to misbehave is strictly smaller.” I believe this is meant to be an explanation for why adversarial training works. I don’t understand the argument well, but in [9], the authors show that adversarially robust models are better for transfer learning than their vanilla counterparts. Intuitively, a broad set of (unique) features must have been learned by the adversarially robust model for this task. I would like to see a discussion comparing these results to [9]. I also think that the analysis is not sufficient to claim that the space of adversarial examples is smaller for R models.

Page 8: “For example, our preliminary results showed that encouraging S networks to have smoother kernels in early layers improves CIFAR-10 ResNet-18 network robustness to adversarial and noisy images.” This has not been shown in this paper/ reference missing.

Suggestion: It would make the work stronger if the authors considered more adversarially robust models, e.g. [10], [11].
Please fix the references: I noticed that the authors cite the arxiv version for [Geirhos et al 2018a]. This paper has been presented at ICLR 2019 and this should be reflected in the references. Please check and fix all other references as well where this critique applies.

References:

[1] Kang et al: “Transfer of adversarial robustness between perturbation types”
[2] Jordan et al: “Quantifying perceptual distortion of adversarial examples”
[3] Tramèr et al: “Adversarial training and robustness for multiple perturbations”.
[4] Tramèr et al: “On Adaptive Attacks to Adversarial Example Defenses”.
[5] Carlini et al: On Evaluating Adversarial Robustness
[6] Rusak et al. “A simple way to make neural networks robust against diverse image corruptions”.
[7] Ford et al: “Adversarial Examples Are a Natural Consequence of Test Error in Noise”
[8] Yin et al: “A Fourier Perspective on Model Robustness in Computer Vision”
[9] Salman et al: “Do Adversarially Robust ImageNet Models Transfer Better?”
[10] Xie et al: “Feature denoising for improving adversarial robustness”
[11] Shafahi et al: “Adversarial training for free!”

---

> ### Author Response · Authors · 2020-11-25
> **Answers to R5 major concerns**
>
> Thank you so much for your encouragement and very detailed, useful feedback!
> We have made changes in light of your questions and constructive criticisms, which, thankfully, made our paper much stronger!
> Please find our inline responses below:
>
> >**I suggest adding and discussing 11 additional publications**
>
> Thank you! In light of your comments, we have added the discussions and citations of all these 11 papers, which we believe strengthens the connection between our work and the literature!
>
>
> >**1. One of their major findings is that adversarially trained models are invariant to additive high frequency noise has been shown and discussed in [8] which they did not cite**
>
> Actually, the performance on additive noise is NOT one of our main findings. Rather, we discovered a series of simplicity biases (shape bias, smooth filters) that EXPLAIN this performance.
> [8] is now discussed in our Sec. 3.3.1 to more strongly motivate our findings. Thanks!
>
>
> >**2. I am not sure how useful and interesting the finding is that adversarially robust models are more shape biased**
>
> Our main contributions are the novel insights into the inner-workings of adversarially robust models and such interesting findings can be in turn harnessed to develop useful, future algorithms. For example:
>
> - We found that R networks heavily rely on shape cues. One may fuse an S network and a R network (two channels, one uses texture and one uses shape) into a single, more robust, interpretable ML model.
> That is, such model may (1) have better generalization on OOD data than S or R network alone and (2) enable an explanation to users on what features a network uses to label a given image.
>
> - We found adversarial training to induce major, consistent internal neural changes to a network. Such findings, e.g. smoother filters (Sec. 3.3.1), neurons detecting lower-level and fewer features (Sec. 3.3.2), can be formulated into a regularizer that can be incorporated into the training of a vanilla network performance (instead of adversarial training which requires ~2x longer training time due to the computation of adversarial examples).
>
> - Our study on how individual hidden neurons contribute to the R network shape preference (Sec. 3.4)  revealed that texture-detector units are equally important to the texture-based and shape-based recognition. This is in contrast to a common hypothesis that texture detectors should be exclusively only useful to texture-biased recognition. This surprising finding suggests that the categories of stimuli in the well-known Network Dissection (Bau et al. 2017; 2019) need to be re-labeled and also extended with low-frequency patterns e.g. single lines or silhouettes in order to more accurately quantify hidden representations.
>
> We have added these points into the Discussion and Future work section.
>
>
> >**3. R networks being shape-biased and having more texture detectors seem contradictory to me and should be discussed.**
>
> Yes! We dedicated further experiments and the entire Sec. 3.4 on this question. That is, we studied how individual neurons (labeled into various categories by NetDissect) are important to shape- or texture-based recognition.
>
> Our explanation for this contradiction is that: adversarial training robustifies networks by simplifying the functions of hidden neurons to detecting more lower-level features.
> That is, R networks have significantly more color and texture detectors (according to NetDissect) than S networks.
> However, these texture detectors are actually useful to both shape- and texture-based recognition (see an example in Fig. 5).
>
> We will make this emphasis stronger in Sec. 3.3 and Sec. 3.4.
>
>
> >**4. The number of PGD steps used for the adversarial attack is 7 which is very low**
>
> We followed exactly the PGD setup (Table 4 in [b]; # step of 7, epsilon of 3, and step size of 0.5) in [a, b, c] to train and test R networks.
>
> Under this PGD setting, R networks were already shown to possess perceptually-aligned hidden representations [a], smooth input-gradients [c], and ResNet-R was shown to serve as a strong image prior [b].
> These works have shown strong evidence of the differences between S and R networks on ResNet-50 and we extended them into 3 different architectures (AlexNet, GoogLeNet, ResNet-50). We added this clarification to the paper.
>
>
> [a] Engstrom et al. 2019 https://arxiv.org/abs/1906.00945
>
> [b] Santukar et al. NeurIPS 2020  https://arxiv.org/abs/1906.09453
>
> [c] Bansal et al. CVPR 2020
> https://openaccess.thecvf.com/content_CVPR_2020/html/Bansal_SAM_The_Sensitivity_of_Attribution_Methods_to_Hyperparameters_CVPR_2020_paper.html

---

> ### Author Response · Authors · 2020-11-25
> **Replies to R5 general comments [1/2]**
>
> >**Page 2 “In contrast, there was also evidence that classifiers trained on one type of adversarial example often do not generalize well to other image types [Geirhos et al, 2018b, Nguyen et al., 2015]”. What is image types here?**
>
> Thanks! We updated the writing to say “...trained on one type of images often do not generalize as well to others”.
> The word “adversarial examples” in this sentence should be “out-of-samples”.
> We’ve also cited your three suggested papers.
>
> >**Table 1: On the torchvision website, the top1 accuracies for the S models are given as: AlexNet: 56.55%, GoogleNet: 69.78%, ResNet50: 76.15%. These numbers differ from what the authors show in Table 1.**
>
> Thanks so much for your catch!
> We have looked into this and found that we had used the pre-processing of transform.resize(256,256) instead of transform.resize(256), causing our numbers to slightly differ.
> We are now able to reproduce 76.13% for ResNet-50 and have updated Table 1 for all networks using the correct pre-processing.
>
>
> >**Page 4: “R models show no generalization boost on ImageNet-C i.e. they performed on-par or worse than the S counterparts (Table 3c). This is consistent with the findings in Table 4 & 5 in Geirhos et al. (2018a) that a stronger shape bias does not necessarily imply better generalizability.” -> This is completely opposite to what we see in Table 5 in Geirhos et al. 2018a.**
>
> We can only partially agree here.
> First, please note that, in Table 4 in Geirhos et al. 2018 (https://arxiv.org/pdf/1811.12231.pdf) , the two best performing networks are “SIN+IN” and “SIN+IN ft IN” are actually TEXTURE-biased (i.e. only shape-based at 34% and 20%; see model_B and model_C in https://github.com/rgeirhos/texture-vs-shape).
> However, the SIN network in Table 4 is SHAPE-biased at 81.37% (https://github.com/rgeirhos/texture-vs-shape/ ) but it has even lower ImageNet-C performance than vanilla ResNet-50 (Table 4).
> This crucial point was missing from their original paper.
>
> Here, we show a consistent result i.e. ResNet-50 trained on adversarial examples (i.e. also non-ImageNet) is shape-biased but has worse ImageNet-C performance than vanilla ResNet-50 trained on ImageNet.
>
>
> >**Page 4: “However, this bias is orthogonal to the performance when only either texture or shape cues are present.” I don’t understand why it would be orthogonal? What exactly is new in this test compared to the cue conflict task?**
>
> First, in cue-conflict images, part of the original texture cue is still present together with the original shape (e.g. see the bird fur pattern in Fig. 1e). Second, the cue-conflict test measures, in network responses, how one cue suppresses the other, which is not necessarily proportional or equivalent to the network response to a single cue alone (i.e. the quantity measured in Sec. 3.2.1 & 3.2.2).
>
>
> >**Page 6: “To confirm this hypothesis”, please write “to test this hypothesis”**
>
> Thank you! We’ve now made this change.
>
>
>
> >**Page 6: “That is, the smooth filters in R models indeed can filter out pixel-wise Gaussian noise despite that R models were not explicitly trained on this image type!” The finding here seems very related to [7] and [8]**
>
> Our finding explains how the result [7, 8] was possible from the network inner functions perspective.
>
>
>
> >**Page 6: “Our result suggests that most of the de-noising effects take place at lower layers (which contain generic features) instead of higher layers.” I do not understand how this conclusion was made given the results, please explain.**
>
> In Sec. 3.3.1, we added Gaussian noise to ImageNet images and compared the layer-wise changes in the total variation of feature maps, before vs. after adding noise. Across all layers, this change is consistently smaller in R networks than that in S networks (Fig. 3). That is, input noise gets propagated to the feature maps of S networks but mostly not in R networks. And the contrast between S vs. R networks appear the largest in lower layers (conv1-conv3) compared to in higher layers (conv4-5).

---

> ### Author Response · Authors · 2020-11-25
> **Replies to R5 general comments [2/2]**
>
> >**Page 6: “We found a consistent trend—adversarial training resulted in substantially more filters that detect colors and textures (i.e. in R models) in exchange for fewer object and part detectors”. This result seems surprising and not consistent with the finding that adversarially robust models are more shape biased. If shape bias is indeed correlated to the filter responses (which is not necessarily clear), I would expect filters of a shape-biased model to rather detect objects and parts instead of textures and colors.**
>
> We wish to clarify that the NetDissect images in objects and parts categories contain BOTH shape and texture cues at the same time (e.g. `dog-object` images contain an entire dog with body and fur). In contrast, the NetDissect images in the texture category contain high-frequency, low-level patterns e.g. stripes or polka dots.
> We found 15% of the units shift from detecting higher-level features (objects, parts) to lower-level features (textures, colors) as the result of changing from standard to adversarial training.
>
>
> >**It would also be nice to test the SIN trained model from Geirhos et al 2018a, since it has a high shape bias.**
>
> Per your request, we added this result to the paper (Sec. 3.3.2). We found that, similar to R models, the SIN model also has fewer object and part neurons and more texture neurons (Fig. A28).
>
>
> >**I would like to see a discussion comparing these results to [9].**
>
> In Sec. 3.3 and 3.4, we found that units in R networks learn more generic and lower-level features, which might be more usable in a new downstream task.
> In contrast, units in S networks are more higher-level and more specialized to ImageNet, which might require longer training to “unlearn” and adapt to new tasks.
>
>
> Page 8: “For example, our preliminary results showed that encouraging S networks to have smoother kernels in early layers improves CIFAR-10 ResNet-18 network robustness to adversarial and noisy images.” This has not been shown in this paper/ reference missing.
>
>
> >**Change the arxiv cite for [Geirhos et al 2018a] to its ICLR 2019**
>
> Thank you! We’ve fixed this for [Geirhos et al. 2018a] and any other applicable arXiv cites in our references.

---

### Decision · Program_Chairs · 2021-01-07
**Final Decision**

**Decision:**

Reject

**Comment:**

This work investigates the relationship between adversarial robustness and shape bias of neural networks. Reviewers pointed out that one of the primary questions being investigated "(a) how adversarially-robust ImageNet classifiers (R classifiers) generalize to out-of-distribution examples;" has already been a primary focus of several prior works, and that many of the findings are already well established, or expected given known connections between adversarial robustness and corruption robustness. I recommend the authors rework the paper to focus more on building upon these prior results. As a possible example, the work would be strengthened if the authors compared adversarial training to other data augmentation strategies known to directly improve shape bias and corruption robustness, does adversarial training provide any unique ood robustness properties distinct from these other methods?